# Material Extrusion Additive Manufacturing of the Composite UAV Used for Search-and-Rescue Missions

**Sebastian-Marian Zaharia** [1,*], **Ionut Stelian Pascariu** [1], **Lucia-Antoneta Chicos** [1], **George Razvan Buican** [1], **Mihai Alin Pop** [2], **Camil Lancea** [1] **and Valentin Marian Stamate** [1]

[1] Department of Manufacturing Engineering, Transilvania University of Brasov, 500036 Brasov, Romania; ionut.pascariu@student.unitbv.ro (I.S.P.); l.chicos@unitbv.ro (L.-A.C.); buican.george@unitbv.ro (G.R.B.); camil@unitbv.ro (C.L.); valentin_s@unitbv.ro (V.M.S.)

[2] Department of Materials Science, Transilvania University of Brasov, 500036 Brasov, Romania; mihai.pop@unitbv.ro

[*] Correspondence: zaharia_sebastian@unitbv.ro

**Abstract:** The additive processes used in the manufacture of components for unmanned aerial vehicles (UAVs), from composite filaments, have an important advantage compared to classical technologies. This study focused on three-dimensional design, preliminary aerodynamic analysis, fabrication and assembly of thermoplastic extruded composite components, flight testing and search-rescue performance of an UAV. The UAV model was designed to have the highest possible structural strength (the fuselage has a structure with stiffening frames and the wing is a tri-spar), but also taking into account the limitations of the thermoplastic extrusion process. From the preliminary aerodynamic analysis of the UAV model, it was found that the maximum lift coefficient of 1.2 and the maximum drag coefficient of 0.06 were obtained at the angle of attack of 12°. After conducting flight tests, it can be stated that the UAV model, with components manufactured by the thermoplastic extrusion process, presented high stability and maneuverability, a wide range of speeds and good aerodynamic characteristics. The lack of this type of aircraft, equipped with electric motors, a traffic management system, and a thermal module designed for search-and-rescue missions, within the additive manufacturing UAV market, validates the uniqueness of the innovation of the UAV model presented in the current paper.

**Keywords:** UAV; additive manufacturing; composite structures; search-and-rescue missions



## 1. Introduction

Unmanned aerial vehicles (UAVs) or so-called aerial drones have experienced a rapid development in recent years, having applications in various industrial fields. The military domain [1–3] has undergone a significant transformation, particularly through the implementation of UAVs in modern warfare strategies, as they can be operated remotely and used for surveillance, reconnaissance, and precise targeting with high accuracy. Currently, UAVs are employed to provide remote medical assistance, offering a new way to deliver supplies and medical services to those in need in remote areas such as rural villages and isolated islands, where traditional delivery methods are not feasible [4–6]. Agricultural drones [7,8] can be used to monitor the condition of crops and animals on farms, map agricultural lands, as well as gather crucial information (crop health, maturity level, moisture level) and perform agricultural tasks like phytosanitary treatments and fertilizer application. Monitoring wildlife using UAV systems provides the opportunity to explore and observe previously inaccessible areas considered challenging or even impossible to reach. UAVs are successfully utilized to observe and document animal behavior, monitor animal migrations, and control poaching activities [9–11]. UAVs equipped with advanced sensors and thermal cameras can be used for [12,13]: scanning large areas in search and rescue operations, monitoring hazardous terrains, investigating disasters, and locating individuals in need

of assistance. Recent uses of UAVs, using thermal vision systems, include the inspection and maintenance of infrastructure such as bridges, tunnels, and wind turbines [14–16]. The delivery process by means of UAVs is implemented in various industries, including the distribution of medical materials (equipment and medicine) and food delivery [17–20].

UAVs designers and manufacturers, when choosing the components materials, consider several factors aimed at functional, aerodynamic, and structural requirements as well as minimizing production and maintenance costs [21,22]. Today, the aerospace research is focused on expanding the use of composite materials in the manufacture of UAVs. Composite materials provide an ideal balance between aircraft weight and aerodynamic strength, fatigue, and corrosion resistance, while reducing maintenance costs [23–26].

To offer a more comprehensive overview of the current research status in this field, five UAV models were studied that shared similar dimensional characteristics and flight performances with the one developed in this study (Table 1).

As a result of this comparative study, the following conclusions were drawn: positioning the wing on the upper section of the fuselage, with a wingspan of 3.4 m, offers numerous advantages (such as easier maintenance, greater stability, a higher lift-to-drag ratio, and improved capacity for gliding over long distances, as well as shorter landing distances); the use of two brushless electric motors, manufactured through metal additive manufacturing, capable of generating a minimum of 11 kg of thrust; implementing a T-tail configuration, with the horizontal tail positioned at 80% of the height of vertical tail, based on considerations including removing the horizontal tail from the wake of the wing, improving structural integrity by connecting the horizontal tail to the vertical tail and fuselage, achieving high stability, and reducing drag; utilizing a tricycle landing gear, which is the most commonly used landing gear configuration, especially for gliders and motor gliders; incorporating a thermal module, allowing the UAV model to perform multiple missions, including search and rescue, surveillance, reconnaissance, and wildlife monitoring. These conclusions represent an initial step in the design, manufacturing, and testing of the UAV model and will serve as input data for the design phases of this study.

**Table 1.** Design parameters and flight performances of UAV with fixed-wing.

| UAV Type | Design Parameters | | Performances Data | |
|---|---|---|---|---|
| Optimum Solutions Condor 300 [27]  | MTOW [kg] | 18 | Cruise Speed [km/h] | 90 |
| | Span [m] | 3.2 | Range [km] | Long range |
| | Wing Area [m²] | N/A | Endurance [hours] | 4 |
| | Type of wing | Low wing | Cruise altitude [m] | 3000 |
| | Tail | T-Tail | Payload [kg] | 6 |
| | Motor type | Twin electric motor mounted on the wings | Mission | Search and Rescue of Missing Persons |
| | Power [W] | N/A | Takeoff Requirements | Autonomous take-off and landing |
| | Battery [mAh] | N/A | Takeoff Distance [m] | 40 |
| Albatross UAV [28]  | MTOW [kg] | 10 | Cruise Speed [km/h] | 68 |
| | Span [m] | 3 | Range [km] | 280 |
| | Wing Area [m²] | 0.683 | Endurance [hours] | 4 |
| | Type of wing | High wing | Cruise altitude [m] | Medium |
| | Tail | Inverted v-tail | Payload [kg] | 4.4 |
| | Motor type | Electric | Mission | Surveillance, search and rescue, reconnaissance |
| | Power [W] | N/A | Takeoff Requirements | Entirely autonomous from takeoff |
| | Battery [mAh] | N/A | Takeoff Distance [m] | 50–100 |

**Table 1.** *Cont.*

| UAV Type | Design Parameters | | Performances Data | |
|---|---|---|---|---|
| Silent Falcon UAS [29]  | MTOW [kg] | 14.5 | Cruise Speed [km/h] | 90 |
| | Span [m] | 4.4 | Range [km] | 15 |
| | Wing Area [m²] | N/A | Endurance [hours] | 5 |
| | Type of wing | High wing | Cruise altitude [m] | 6000 |
| | Tail | Conventional cruciform | Payload [kg] | 3 |
| | Motor type | 1.3-hp electric motor | Mission | Search and rescue, wildlife monitoring, agricultural survey |
| | Power [W] | N/A | Takeoff Requirements | N/A |
| | Battery [mAh] | N/A | Takeoff Distance [m] | N/A |
| Vector VTOL fixed wing UAS [30]  | MTOW [kg] | 7.4 | Cruise Speed [km/h] | 70 |
| | Span [m] | 2.8 | Range [km] | 25 |
| | Wing Area [m²] | N/A | Endurance [hours] | 2 |
| | Type of wing | High wing | Cruise altitude [m] | N/A |
| | Tail | T tail | Payload [kg] | 0.4 |
| | Motor type | N/A | Mission | Search and rescue, convoy protection, border patrol, traffic investigation |
| | Power [W] | N/A | Takeoff Requirements | Vertical takeoff |
| | Battery [mAh] | N/A | Takeoff Distance [m] | N/A |
| Penguin BE UAV [31]  | MTOW [kg] | 21.5 | Cruise Speed [km/h] | 79.2 |
| | Span [m] | 3.3 | Range [km] | 180 |
| | Wing Area [m²] | 0.79 | Endurance [hours] | 2 |
| | Type of wing | High wing | Cruise altitude [m] | 6000 |
| | Tail | V-tail splits in two parts | Payload [kg] | 6.6 |
| | Motor type | Gas/Electric engine | Mission | N/A |
| | Power [W] | 2700 | Takeoff Requirements | Runway, catapult or car-launched |
| | Battery [mAh] | N/A | Takeoff Distance [m] | 30 |

A modern process for manufacturing components and prototypes of UAVs is the additive process of thermoplastic material extrusion (filament). The additive manufacturing process by materials extrusion provides the following advantages: low price of equipment and materials used, wide range of materials, high manufacturing speed, freedom and creativity on the design of the part, user-friendly interface [32,33]. The additive thermoplastic material extrusion process has two similar terms in the specialized literature [34–36]: FFF (Fused Filament Fabrication) and FDM™ (Fused Deposition Modeling) which describes the same type of additive 3D printing process in which material (filament, fused material) is selectively extruded through a nozzle or orifice.

However, components manufactured by additive material extrusion processes show lower mechanical properties, compared to classic composite materials. This inconvenience comes from the limited properties of the matrices, but also because of the low adhesion between the layers of extruded material. These defects cause a decrease in the strength and elongation of the 3D-printed parts [37,38]. The evolution of 3D-printing systems, by controlling the temperature of the working area, has led to the use of high-performance thermoplastic materials such as [39–41]: Polyether Ether Ketone (PEEK), Polyetherimide (PEI), and composite materials (reinforced with carbon and glass fiber).

Additive manufacturing processes by material extrusion have begun to be used for the realization of UAV prototypes that are tested in flight [42,43]. Starting from three-dimensional digital models to flying UAV prototypes, this process is completed in a few days with minimal designer intervention. Another direction of study is represented by the additive manufacturing by material extrusion of drones [44], starting from the testing stages

of lightweight cellular structures [45] or by using topological optimization techniques for the resistance structure [46].

For the UAV models, produced by material extrusion, the aerodynamic performances were determined by testing in the wind tunnel [47–49]. The manufacturing of morphing wing models [50–53] lends itself very well to the additive manufacturing process by material extrusion, since the components are manufactured in the shortest time and at low costs. Currently, the use of composite filaments, with short [54–56] or continuous fibers [57,58], employed in the 3D-printing process represents an intensively researched field with applications in aerospace. Recent studies have highlighted that by adding short carbon fibers to a PAHT (High-Temperature Polyamide) matrix [59–61] or short glass fibers to a PLA (Polylactic acid) matrix [62–64], the mechanical performance of manufactured parts exhibited a significant increase in various tests (compression, impact, traction, flexural). While various types of UAVs have been manufactured from standard materials (PLA, ABS, PETG) until now, this study extended research to two types of materials reinforced with short carbon and glass fibers, with the goal of enhancing the mechanical performance of UAV model components. Moreover, through the use of these two types of short fiber-reinforced materials, the UAV model will exhibit strong structural strength, leading to higher flight safety.

In this paper, the ability of the additive material extrusion processes of filament reinforced with short carbon and glass fiber to obtain an unmanned aerial vehicle was demonstrated. After completing the stages of design, preliminary aerodynamic analysis, 3D printing, and assembly of the UAV model, the flight tests were performed and the search-and-rescue mission was accomplished using a thermal camera module.

## 2. Design of UAV Model

### 2.1. Preliminary Design

For the preliminary design of the composite UAV, four important parameters must be established [65,66]: maximum takeoff weight (MUAV), minimum wing area (Smin), motor thrust (T), and autopilot preliminary calculations. In the design process of the UAV model, special attention was paid both to the parameters that must be minimized ($M_{UAV}$) and the performances that must be maximized within the constraints (autonomy, flight ceiling, aerodynamic performance). In the first stage of designing the UAV model, the basic configuration for the four important structural components (wing, fuselage, wings, and control surfaces) was established.

Throughout the design and physical realization processes of the UAV model, several steps were followed, as presented in Figure 1 [67]. These stages commence with the requirements stage and mission profile establishment, progressing through conceptual design, preliminary design, and detailed design. The UAV prototype was developed to facilitate additive manufacturing of components, followed by testing and flight mission completion. The design and physical realization of the UAV model is a creative and iterative process that combines technical knowledge, experience, and problem-solving skills to develop the UAV solution to answer real-world problems. The seven stages of UAV model development are detailed below.

1. Following the design process depicted in Figure 1, the starting point for the UAV model is to comprehend its purpose and flight mission to determine the appropriate design requirements for the aircraft. Since this study involved design and physical creation of a fixed-wing, twin-engine aircraft for search-and-rescue missions utilizing a thermal module controlled by a ground control station, both flight mission planning and design requirements had to be individually satisfied and concurrently addressed [67]. The UAV model is intended for use in mountainous areas to locate pilots and passengers after aviation accidents, wildlife tracking, and detecting illegal hunting. The selection of these missions for the UAV model was not arbitrary; it was based on studies that highlight significant deficiencies in finding economically and technically efficient solutions for aviation accident response and anti-poaching efforts in mountainous regions.

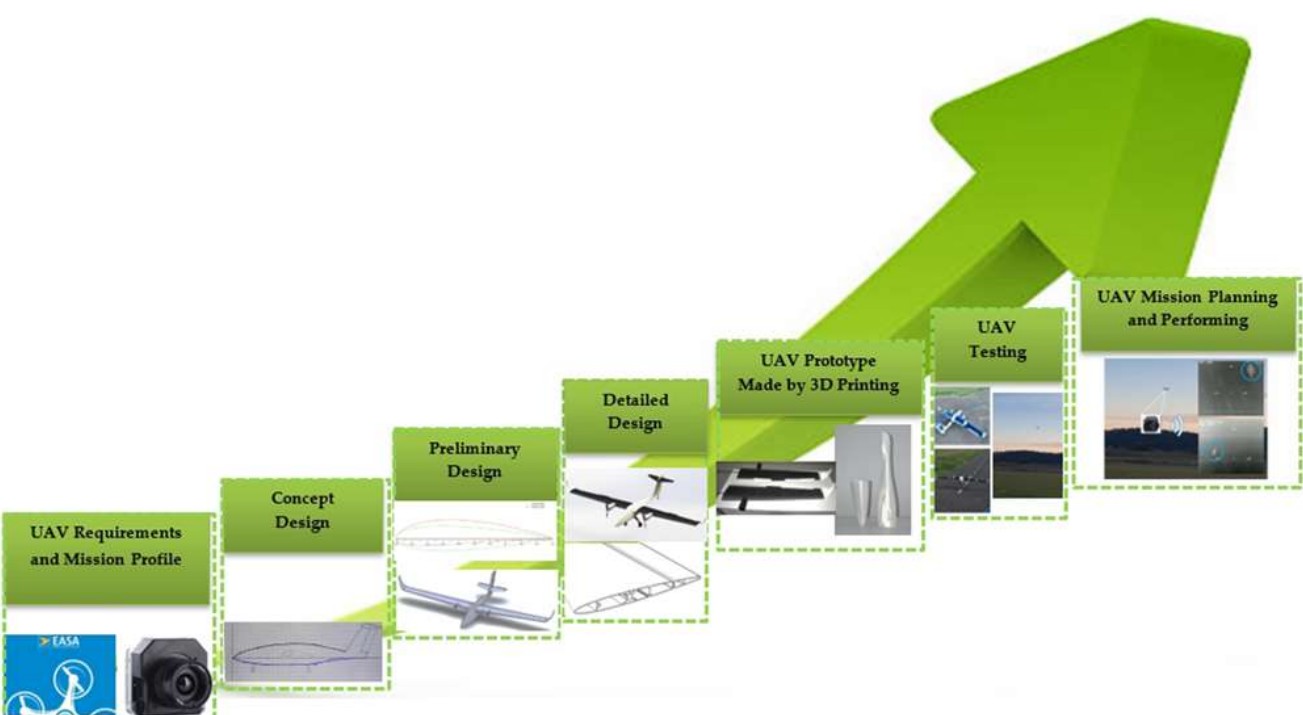

**Figure 1.** The engineering design process of the UAV model.

2. The conceptual design stage of the UAV model involves analyzing current concepts and designs based on the defined design requirements to visualize the desired UAV configuration. This initial design investigation is crucial, since the entire design phase, up to detailed design, relies on this preliminary analysis [68] and the proposed concept. During this stage, the project team engages in brainstorming sessions that result in a predimensioned sketch of the UAV model. These sessions evaluate the advantages and disadvantages of each proposed idea, taking into consideration specifications, flight mission, manufacturing costs, and the fabrication process [69].

3. In the preliminary design stage, the initial UAV model will be expanded and developed in much greater detail. Optimization compromises within the project will be made to maximize the performances of the aircraft for its intended operational roles and flight mission. In this stage, decisions will be made about which UAV components will be fabricated in-house and which ones will be purchased, involving a cost–benefit analysis. Utilizing specific Computer Aided Design (CAD) software systems, the preliminary UAV model was designed based on the previously established basic dimensions. Additionally, in this stage, estimates of the masses of the main components will be made from the CAD model of the aircraft, along with establishing aerodynamic profiles (for the wing and empennage) and obtaining initial results regarding the aerodynamic performance of the UAV model [70].

4. In the subsequent detailed design stage [71] of the UAV model, computerized methods for design, calculations, simulations (virtual development), and detailed aerodynamic analyses (wing analysis, aircraft analysis) were employed. Furthermore, full characterization (mechanical and thermal) of the two types of materials used in 3D printing the UAV model [72,73] was performed in this stage. The detailed design for the additive manufacturing of the UAV components was also included, considering the 3D printing volume of the equipment used and the method of support structure application and removal for certain parts. In this stage, aspects of performance (autonomy, flight speed, ceiling) of the UAV model were obtained as the components to be used on the aircraft (electric motors, ESC, battery) were determined.

5. During the 3D printing phase of the UAV prototype, detailed CAD models and manufacturing parameters from the previous stage were used to produce segments of

all UAV model components. An important aspect of this phase was testing the mechanical performance of the wing [74], fuselage [73], and landing gear [75]. Following mechanical testing, the wing and fuselage proved capable of withstanding flight loads; however, a solution in the form of CFRP material was chosen for the landing gear to ensure safer takeoff and landing. Also at this stage, detailed tests were carried out on the motors manufactured through the SLS (Selective Laser Sintering) process [76] to determine their performance. This stage concludes with the simultaneous assembly of aircraft components and electronic elements, resulting in the final physical UAV model.

6.  Within the testing stage, initial ground tests of the UAV model were conducted, monitoring the correct functioning of controls, motors, and electronic systems (thermal module and ground control station). Ground rolling tests of the UAV model were also conducted during this testing stage. Following these preliminary tests, the final stage involved testing the UAV model in flight using the ground control station and thermal module.

7.  The final stage in the development of the UAV model included the flight mission, namely search and rescue using the thermal module. At this stage, the proposed UAV model was validated by successfully completing the specified mission outlined during the flight mission profile establishment phase. The fixed-wing UAV model with a medium altitude is capable of performing search and rescue, surveillance missions, and emergency interventions, serving as a versatile and efficient solution with reduced operating costs. Moreover, the UAV model offers an approximate flight time of 50 min and can cover large search areas rapidly (around 100 km) by thermal imaging capture. Another important aspect of the UAV model is its real-time video transmission capability, providing rescue teams with a clear and detailed view of the operational area, enabling rapid decision-making and a timely response.

*2.2. UAV Model Wing Design*

The first two design characteristics established for the UAV model were the following: maximum weight of about 11 kg and median positioning of the wings on the fuselage. The minimum wing area ($S_{\min}$) for the UAV model [42] was calculated using Equation (1) with the following parameters: maximum takeoff weight of the UAV model ($M_{UAV}$ = 11 kg), gravitational acceleration ($g$ = 9.81 m/s$^2$), air density at sea level ($\rho$ = 1.225 kg/m$^3$), maximum lift coefficient ($C_{L\max}$ = 1.2), and minimum airspeed of the UAV ($V_{\min}$ = 13 m/s).

$$S_{\min} = \frac{2 \cdot M_{UAV} \cdot g}{\delta \cdot C_{L\max} \cdot (V_{\min})^2}, \tag{1}$$

The minimum wing area of the UAV model has been calculated to be 0.868 m$^2$. To initiate the wing design process of the UAV model, the minimum wing area required for the aircraft's flight was established. The calculated minimum wing area using Equation (1) was 0.868 m$^2$, which is smaller than the wing area of the UAV model (0.897 m$^2$).

Starting from the minimum surface determined with Equation (1), the UAV model will have a wingspan (b) of 3400 mm and the wing will be positioned in the upper part of the fuselage, which determines good stability and reduces the possibility of the wing the ground. To determine the dimensions of the UAV wings, a new constraint is necessary. Tapering the wing provides significant aerodynamic advantages (reducing drag force and increasing lift distribution) as well as structural benefits (weight savings for the wing). In the case of UAVs, this taper ratio typically has a value of 0.5 [77,78], a value obtained from data of UAV's with similar configurations and missions. However, for the UAV model analyzed in this paper, a higher value ($\lambda$ = 0.57) was chosen because a winglet will be mounted at the tip of wing. This taper ratio is necessary, as the wing-tip chord needs to provide a larger surface area to facilitate the winglet connection and achieve a higher

strength. The root chord of the UAV model was calculated using Equation (2) [77,78], while the tip chord was determined using Equation (3) [77,78].

$$C_r = \frac{2 \cdot S}{b \cdot (\lambda + 1)}, \tag{2}$$

$$C_t = 0.57 \cdot C_r, \tag{3}$$

The mean aerodynamic chord (MAC) of the UAV model was computed using Equation (4) [77,78].

$$MAC = \frac{2}{3} \cdot C_r \cdot \frac{1 + \lambda + \lambda^2}{1 + \lambda}, \tag{4}$$

The aspect ratio [77,78] of the UAV model represents the ratio between the squared wingspan and the wing area (Equation (5)).

$$AR = \frac{b^2}{S}, \tag{5}$$

Another important parameter of the UAV model is the wing loading (*WL*), which was determined using Equation (6) [77,78].

$$WL = \frac{M_{UAV}}{S}, \tag{6}$$

The wing area ($S$ = 0.897 m$^2$) and wing loading ($WL$ = 120 N/m$^2$) are currently the most significant constraints for wing sizing and design. The wing used for the aircraft was trapezoidal with a NACA 4415 airfoil. Other basic dimensions of the UAV model were: root chord ($C_r$) 335 mm and tip chord ($C_t$) 200 mm (calculated with Equations (2) and (3)). The design of the wings started by using the same airfoil (NACA 4415) at the root and tip of wing. The NACA 4415 airfoil has a relative thickness of 15% and an optimum value for obtaining a maximum lift coefficient. The design of the UAV model was made in the SolidWorks 2021 software system (Dassault Systèmes SolidWorks Corporation, Waltham, MA, USA).

The two electric motors nacelles were also attached to the wing structure. The structural stiffening of the wing has been carried out using a carbon fiber tubular spar (Figure 2a). The internal configuration of the wing shows a three-spar structure (Figure 2b), as follows: a C-shaped spar at the leading edge of the wing, a X truss type spar that is intended to take the stresses from the central part of the wing and offer a mounting platform the carbon fiber tubular spar. The C-shaped spar is required to take the stresses from the wing trailing edge area and has built in cylindrical surfaces, through which the carbon rods needed to facilitate the guidance of the 3D printed wing sections will be inserted. The position of the wing spars is in accordance with aeronautical structure practice: the first C-shaped spar is positioned between 17 and 25% of the chord, the second X truss type spar is placed between 30 and 45% of the chord, and the third C-shaped spar is placed between 60 and 75% of the chord. Other dimensions of the UAV model wing were the following: the thickness of the wing skin was 1 mm; the thickness of the three spars was 0.8 mm; the diameter of the tubular spar was 16 mm.

Also, in order to reduce the mass of the UAV model and facilitate manufacturing, it was chosen to place the resistance ribs on the bonding surfaces between the 3D-printed wing sections, by adding 3 mm borders. The support for the electric motors has been designed on the wing for a higher resistance, because this area is of vital importance in the operation of the UAV model. To control the roll axis, the wing was equipped with ailerons (Figure 3a), which has a resistance structure consisting of a longitudinal spar with I-profile.

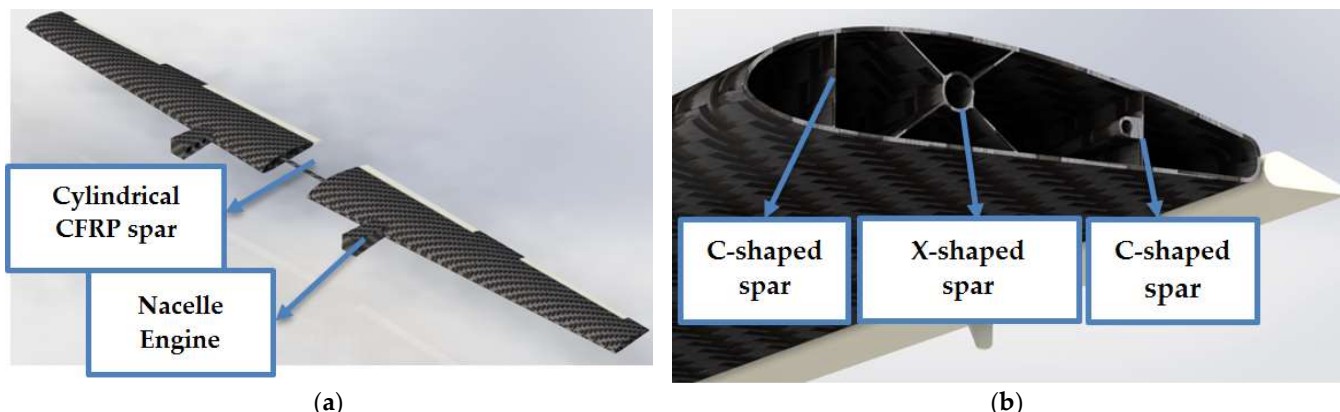

**Figure 2.** UAV model wing design: (**a**) wing with tubular spar; (**b**) the structural components of the wing.

The hyper-sustentation devices used for the UAV model aircraft are the flaps (Figure 3a). The flaps were arranged on the trailing edge of the wing, and by turning them, the lift will be increased during the takeoff and landing stages. The winglet (Figure 3b) reduces the induced drag at the wingtips and increases the lift to drag ratio. By using winglet devices, the power consumption is reduced and the performance of the electric motors used on the UAV model is increased. These engineered components will have a wing-like structure with spars and ribs to facilitate 3D printing without support material and to provide a structure with high stiffness.

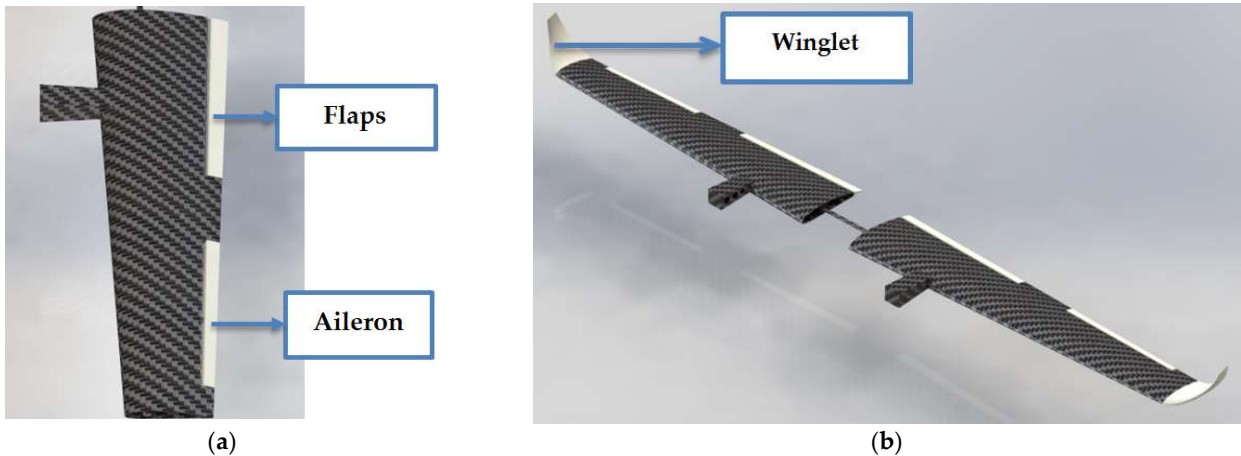

**Figure 3.** Wing structure: (**a**) wing equipped with flaps and aileron; (**b**) wing equipped with winglet.

### 2.3. UAV Model Fuselage Design

The fuselage of the UAV model (Figure 4a) has an aerodynamic shape and is the component to which the following structures are attached: the wing, the horizontal and vertical empennage, the landing gear, and the thermal vision camera. In order to provide the best resistance to the model, the junction of the wing and the fuselage will be manufactured, as a single part, by thermo-plastic extrusion. The aerodynamic shape of the fuselage ensures the maximum load capacity, at the lowest forward resistance.

To determine the fuselage length of the UAV model, two constraints were employed: it should be between four and six times the mean aerodynamic chord and half of the wingspan. Considering these constraints, the fuselage length ($Lf$) can be estimated using the UAV wingspan ($b$), as provided in Equation (7) [78].

$$\frac{Lf}{b} = 0.53, \tag{7}$$

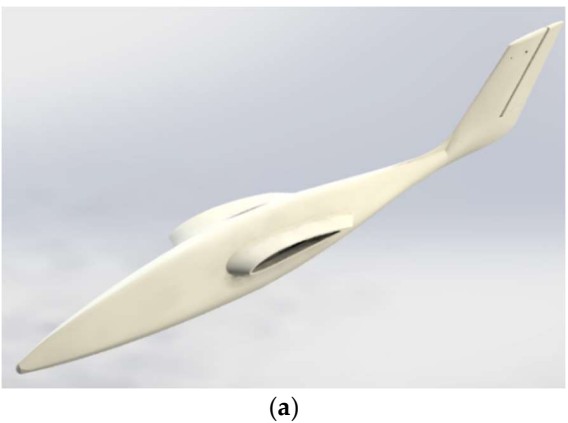
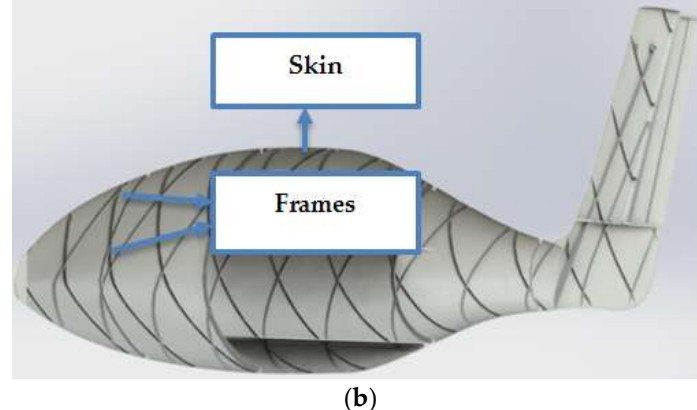

(**a**)                                           (**b**)

**Figure 4.** Structure of the fuselage: (**a**) 3D model of the fuselage and vertical empennage; (**b**) fuselage strength structure.

In this study, particular value of the fuselage length-to-wingspan ratio was chosen because the vertical tail configuration shares a common area with the fuselage in the tail landing gear section of the UAV model. As a result, the calculated fuselage length using Equation (6) is 1.8 m.

According to statistical data for subsonic aircraft, a practical range for the fineness ratio (Equation (8)) falls between 6 and 8, which is considered optimal [78–80]. However, this can vary based on fuselage design requirements and the placement of electronic equipment and engines. Therefore, in this study, a fineness ratio ($\lambda f$) of 8 was utilized for the design phases. An optimum fineness ratio of the UAV fuselage corresponds to minimal drag force.

$$\lambda f = \frac{Lf}{Dmean}, \tag{8}$$

where $Lf$ is the length of the fuselage (m) and $Dmean$ is the mean diameter of the fuselage (m). Consequently, the mean diameter of the UAV model will have a value of 225 mm.

The electronic equipment will be positioned as close as possible to the center of gravity of the UAV model, so that the moments of inertia are as low as possible. The fuselage will have cutouts that will allow access to the electronic components. Also, reinforcement elements will be added to the fuselage structure in the areas where the stresses are more intense (the attachment of the landing gear to the fuselage, the attachment of the thermal chamber to the fuselage).

The fuselage of the UAV model (Figure 4b) is a monocoque type structure, with components of shell and frames type. The frames are positioned in two directions at a 45° angle in order to facilitate a support-free 3D printing. The main dimensions of the fuselage components are the following: thickness skin of fuselage, 1 mm; stringers oriented in two directions at an angle of ±45° (X stringers); stringers thickness, 1 mm and stringers height, 4 mm. This constructive solution was chosen to use the interior space of the fuselage as efficiently as possible, and to facilitate its assembly and manufacture.

*2.4. Design of Empennages and Landing Gear*

In Figure 5a, it can be seen that the fuselage and a part of the vertical empennage were designed together. A classical configuration was selected for both the horizontal and vertical empennage, comprising of a fixed and a movable surface. Thus, the horizontal empennage presents horizontal stabilizer and elevator, and the vertical empennage presents vertical stabilizer and ruder. The horizontal empennage was positioned in the upper section of the rudder, to isolate it from the wing's effects. The airfoil, for both empennages, was NACA 0015. The NACA 0015 profile is a symmetric and thin airfoil used to maintain the

order of magnitude of lift values during elevator or rudder command while also providing minimal drag resistance [70].

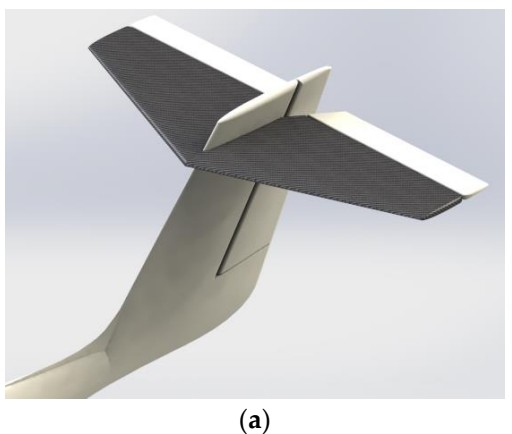 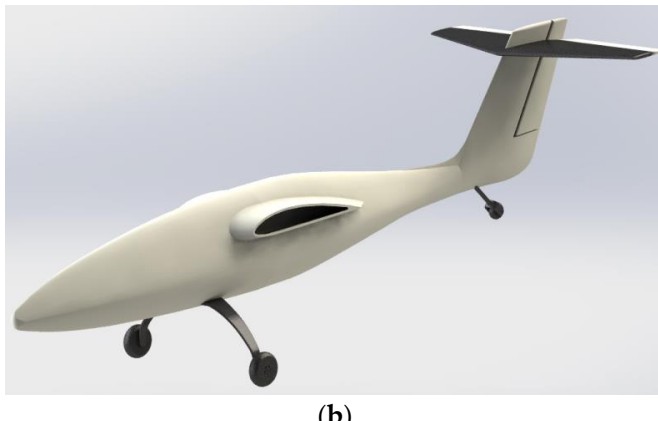

(**a**)　　　　　　　　　　　　　　　　　　　　　　(**b**)

**Figure 5.** Design of the components of the UAV model: (**a**) horizontal and vertical empennage; (**b**) fuselage assembly—empennages—tricycle landing gear.

In this stage of the UAV model design process, the tail surfaces (horizontal and vertical) were estimated using empirically found tail volume coefficients [79–81], as presented in Equations (9) and (10) [81,82]. For an aircraft with two engines mounted on the wing, like the UAV model, the horizontal tail moment arm ($l_{HT}$) was set to 1.25 m, representing the distance between the center of gravity and the aerodynamic center of the horizontal and vertical tail surfaces. Considering a horizontal tail volume ratio ($V_{HT}$) of 0.5 [81,82] for a UAV-type aircraft, horizontal tail wing area ($S_{HT}$) can be determined using Equation (9).

$$V_{HT} = \frac{l_{HT} \cdot S_{HT}}{MAC \cdot S}, \tag{9}$$

Therefore, the horizontal tail wing area ($S_{HT}$), calculated using Equation (9), for the UAV model was 0.098 m². For the UAV model, the vertical tail moment arm ($l_{VT}$) was 1.1 m, and the vertical tail volume ratio ($V_{VT}$) was set at 0.2 [82,83].

$$V_{VT} = \frac{l_{VT} \cdot S_{VT}}{b \cdot S}, \tag{10}$$

The vertical tail wing area ($S_{VT}$), calculated using Equation (10), has a value of 0.055, and the dimensions of the UAV model, computed using Equation (1) to Equation (10), have been presented in Table 2.

**Table 2.** Characteristics of the UAV model.

| Component | Geometric Characteristic | Value |
|---|---|---|
| | Wingspan [m] | 3.4 |
| | Wing area [m²] | 0.897 |
| | Root chord [m] | 0.335 |
| Wing | Tip chord [m] | 0.200 |
| | Taper ratio | 0.597 |
| | Aspect ratio | 12.887 |
| | Mean aerodynamic chord [m] | 0.273 |
| Flaps | Chord [m] | 0.05 |
| | Span [m] | 0.6 |
| Aileron | Chord [m] | 0.045 |
| | Span [m] | 0.65 |

**Table 2.** *Cont.*

| Component | Geometric Characteristic | Value |
|---|---|---|
| Winglet | Root chord [m] | 0.2 |
| | Tip chord [m] | 0.054 |
| | Height [m] | 0.18 |
| Fuselage | Length [m] | 1.803 |
| | Height [m] | 0.228 |
| | Width [m] | 0.200 |
| Vertical tail | Span [m] | 0.420 |
| | Root chord [m] | 0.285 |
| | Tip chord [m] | 0.200 |
| Horizontal tail | Span [m] | 0.600 |
| | Root chord [m] | 0.235 |
| | Tip chord [m] | 0.120 |
| Landing gear | Height [m] | 0.145 |
| | Wheel Track [m] | 0.315 |

For the manufacture of the empennages, the following strength structures were chosen: the rudder contained ribs positioned at 45°, the rudder included a 60° positioned I-beam spar, the stabilizer featured the wing's structural design (three spars), and the elevator had an I-beam spar. The landing gear of the UAV model aircraft will make it possible to taxi safely on the ground without damaging the aircraft during taxiing, takeoff, and landing. The chosen landing gear configuration for this aircraft is a non-retractable tricycle landing gear, consisting of two side arms and a tail wheel (Figure 5b). The main legs of the landing gear were positioned in the wing's trailing edge area.

*2.5. Digital Assembly of the UAV Model*

The assembly of the UAV model involved the interconnection of all the essential components (wing, winglet, aileron, flaps, fuselage, landing gear, vertical and horizontal empennage, electric motors, and thermal imaging camera) to showcase the preliminary design. Components assembly was performed using SolidWorks 2021 software system, using the constraints necessary to build the digital model of the UAV. During this stage, the UAV model components were checked for any potential collisions, ensuring a smooth additive manufacturing process without any issues. The assembly process proceeded as follows: it started with the fuselage structure, onto which the left half-wing and right half-wing were subsequently mounted. The left wing and right wing were equipped with ailerons, flaps, winglets, and electric motors. The assembly procedure continued with the positioning of the horizontal and vertical empennages and was finished with the addition of the tricycle landing gear and a thermal imaging camera (Figure 6).

Table 2 summarizes the main characteristics of the UAV model resulting from the preliminary design. It can be seen that the values of the control surfaces (aileron, elevator, and rudder) fall within the range of typical values of airplanes, which are found in various studies in the field [84,85].

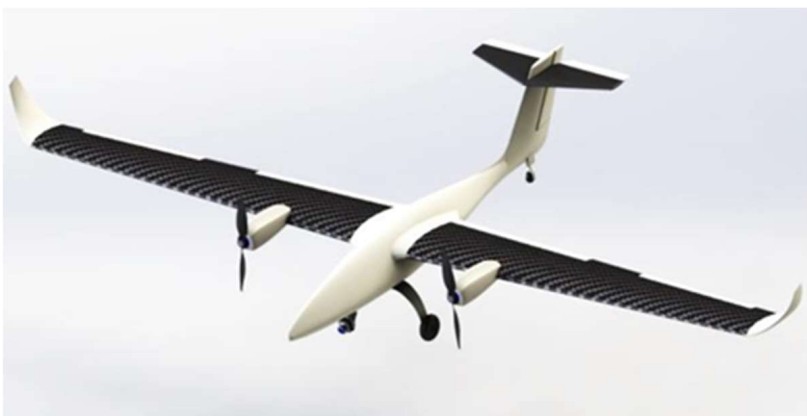

**Figure 6.** Digitally assembled UAV model.

## 3. Preliminary Aerodynamic Analysis

The XFLR5 software system represents a tool used for the analysis of aerodynamic profiles, wings, and the entire aircraft, aiming to determine the aerodynamic performance in the early stages of the lifecycle of an aeronautical product [86,87]. The modeling of the UAV model was carried out using the geometric characteristics from Table 2. The main stages of the modeling process were as follows: importing the aerodynamic profiles (Figure 7a) NACA 4415 (used for the wing) and NACA 0015 (used for the empennages and winglet), modeling the wing and winglet of the aircraft (Figure 7b), modeling the fuselage and empennages (Figure 7c). The final stage involves the assembly of the UAV model (Figure 7d) by constraining the previously modeled components.

The methodology for calculating the aerodynamic performance of the UAV model, using the XFLR5 software system, is described in Figure 8a. After selecting the aerodynamic profiles, the second step in the aerodynamic analysis of the UAV model was to determine the Reynolds number. The kinematic viscosity ($\nu$) was calculated using the following parameters: altitude of 300 m; temperature of 20 °C, resulting in $\nu = 1.647 \cdot 10^{-5}$ m$^2$/s. Using the kinematic viscosity of $1.647 \cdot 10^{-5}$ m$^2$/s, a velocity of 20 m/s, and an aerodynamic mean chord of 0.273 m, the Reynolds number was calculated to be Re = 333,966. The aerodynamic analysis of the UAV model was carried out for a variation of the attack angle within the range of $-5°$ to $12°$. Figure 8b shows the variation of the pressure coefficient at an attack angle of $5°$.

Figure 8c depicts the variation of the lift coefficient for the entire aircraft as a function of the attack angle. As can be seen, the maximum lift coefficient ($C_L$) reached a value of 1.2 at an attack angle (Alpha) of $12°$. Figure 8d shows that the drag coefficient ($C_D$) increased with increasing angle of attack (Alpha) and reached a value of 0.06 at an attack angle of $12°$.

The variation of aerodynamic coefficients is important for flight tests because the obtained results describe the first aerodynamic performances of the UAV model. To evaluate the aerodynamic performance of the UAV model, the values of the lift coefficient ($C_L$) and the values of the drag coefficient ($C_D$) must be known at each angle of attack—in widespread use is the method by which the aerodynamic characteristics are represented in the form of a graph called polar of the airplane.

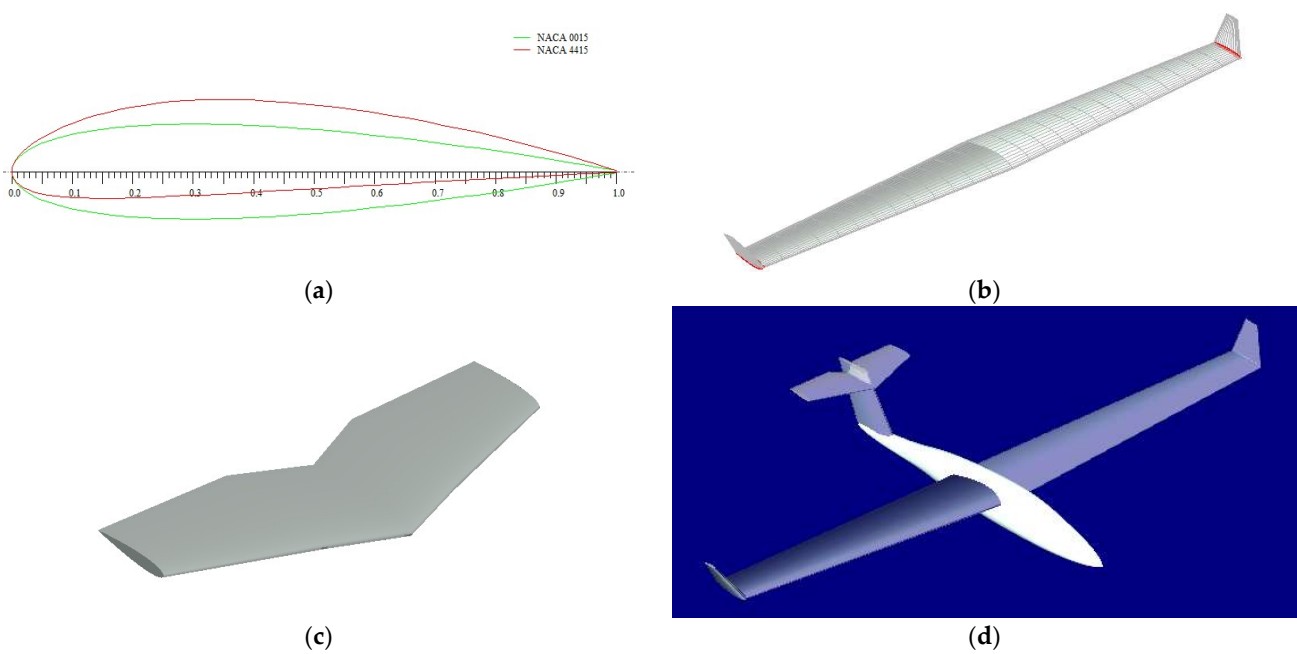

**Figure 7.** Modeling of the UAV model: (**a**) aerodynamic profiles used; (**b**) modeling the wing equipped with a winglet; (**c**) modeling the horizontal empennage; (**d**) UAV model made in XFLR5 software system.

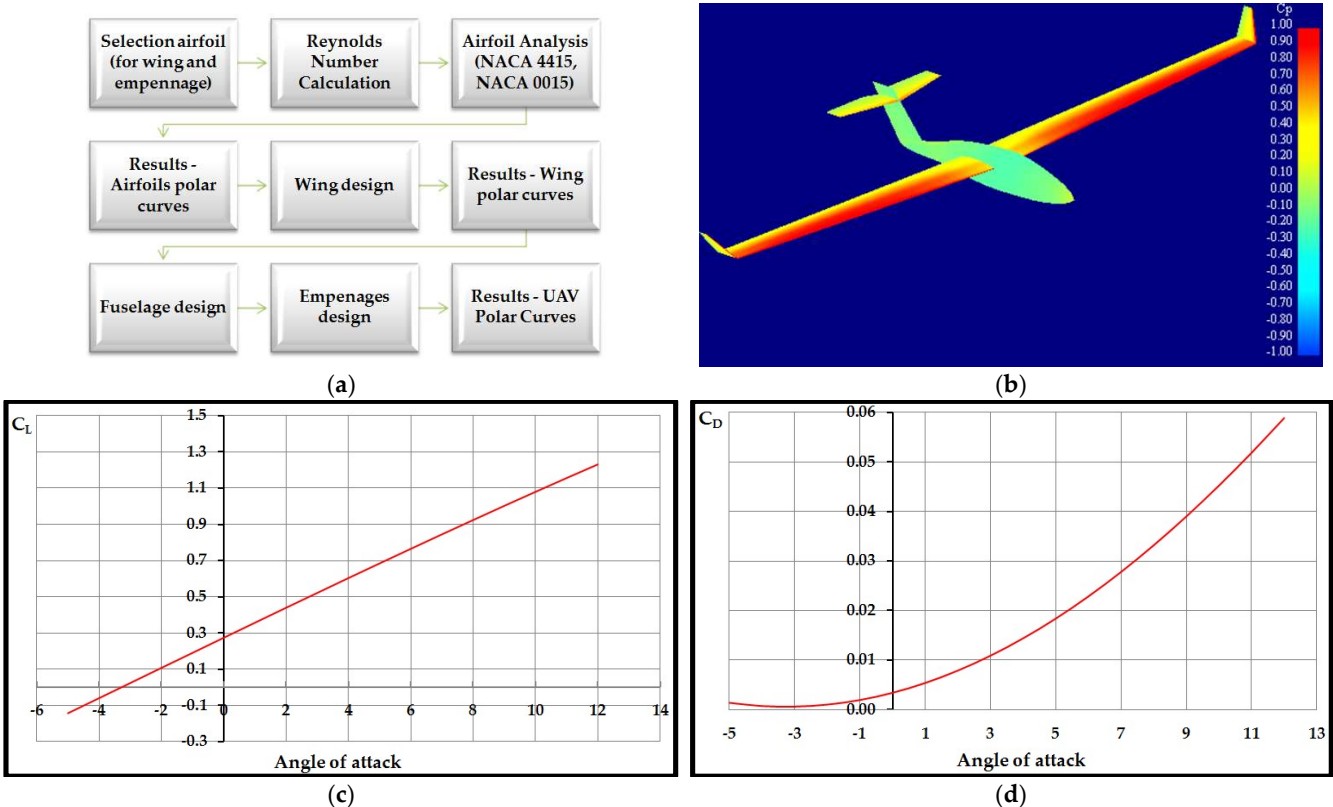

**Figure 8.** Aerodynamic analysis of the UAV model: (**a**) calculation methodology in XFLR5; (**b**) pressure coefficient distribution on the UAV model (angle of attack 5°); (**c**) dependence of lift coefficient ($C_L$) and angle of attack; (**d**) dependence of drag coefficient ($C_D$) on angle of attack.

## 4. Manufacturing of the UAV Model Components Using FFF Process

### 4.1. Fabrication of the Wing Structure Using the FFF Process

This activity began with the testing and manufacturing of specimens and components of the aircraft using various carbon fiber filaments to determine the mechanical performance and optimal manufacturing parameters of the FFF process. In this regard, several studies have been published regarding the FFF manufacturing of specimens/components of the UAV model [73,74]. The first phase in aircraft manufacturing involved dividing the UAV model (Figure 9a) into simpler components suitable for FFF manufacturing using the Ultimaker S5 printer (Ultimaker, Utrecht, The Netherlands).

To manufacture the wings, they were segmented because of their dimensions exceeded the print volume of the printer. Thus, each half-plane of the UAV wing structure was divided into 12 segments (Figure 9b) manufactured as follows: wing segments made from filament reinforced with short carbon fibers—PAHT CF15 [88]; aileron, flap, and winglet segments made from filament reinforced with short glass fibers—Filament PLA Glass Reinforced [89].

The PAHT-CF 15 filament is a high-temperature polyamide reinforced with 15% carbon fiber, exhibiting the following characteristics: high chemical resistance, resistance to high temperatures up to 150 °C, capability to manufacture strong and rigid components, high dimensional stability, ease of processing. The PAHT-CF 15 filament can be successfully used in aerospace and automotive applications [88], increasing the lifespan of components manufactured from this material.

The PAHT-CF15 filament was characterized by [72] in terms of mechanical properties (tensile tests and three-point bending) by varying the infill density (25%, 50%, 75%, and 100%), as well as infill patterns [90]. Once again, it was demonstrated that the specimens with 100% infill density exhibited the highest values for both tensile strength (90.8 MPa) and bending strength (114 MPa). Following the tests [90], for the same type of material (PAHT-CF15), the triangles pattern showed the highest values in three-point bending (108.2 MPa).

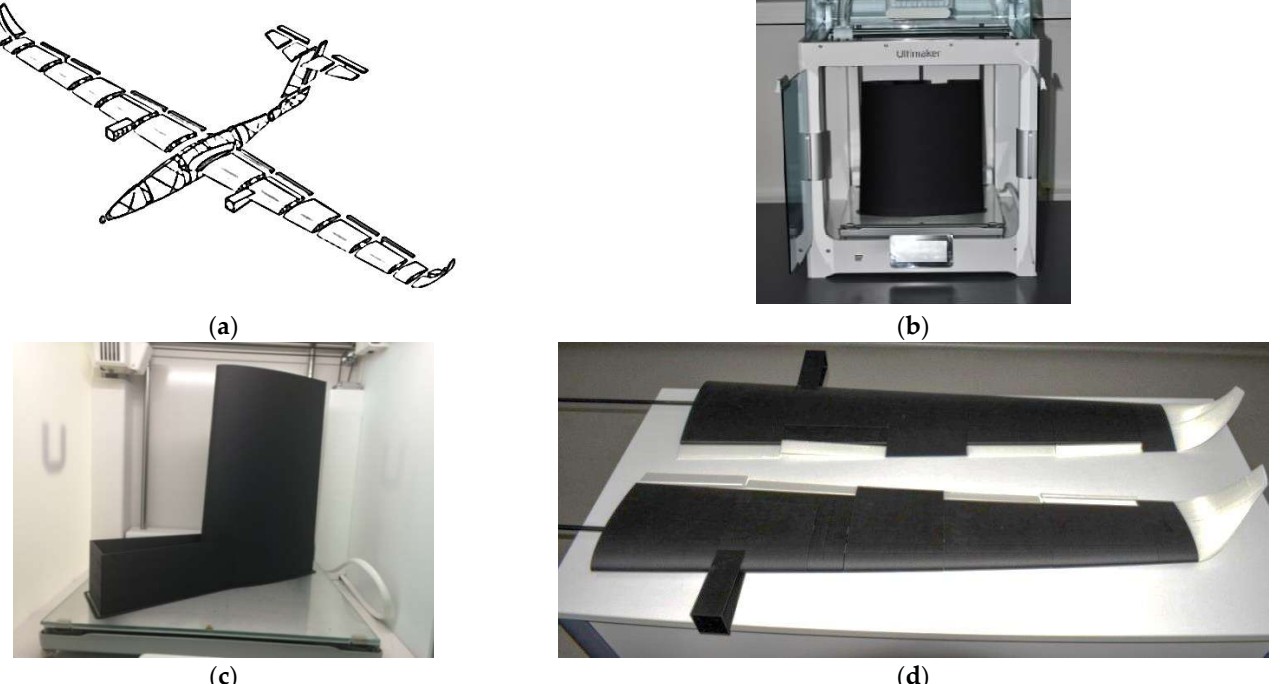

(**a**)　　(**b**)

(**c**)　　(**d**)

**Figure 9.** Manufacturing the UAV using the FFF process: (**a**) the UAV model prepared for FFF manufacturing; (**b**) manufacturing of wing segment 4; (**c**) manufacturing of segment 2, including the engine mount; (**d**) left half-plane and right half-plane of the UAV wing.

The PLA glass-reinforced filament is a biopolymer that offers superior strength and hardness compared to standard PLA filament, making it suitable for FFF-based manufacturing of lightweight components in the aviation and automotive sectors, with reduced warping [89].

For enhanced flight safety, the selection and testing of materials used in the construction of the aircraft components represented a crucial stage in the design and physical realization of the UAV model. Consequently, the testing of PLA glass material was carried out under tensile, flexural, and compressive loading conditions, with the following results: a tensile strength of 49.6 MPa, a flexural strength of 89 MPa, and a compressive strength of 68.2 MPa were obtained for specimens with 100% infill density [73]. Among the three types of tests conducted, it was evident that specimens with a 100% infill density exhibited the highest mechanical characteristics, despite the longer 3D printing time and higher material consumption compared to lower infill density cases of 75% or 50% [73].

After manufacturing the wing segments (Figure 9b,c) for the right half-plane, the next step involved fabricating the segments for the left half-plane (Figure 9d). For manufacturing preparation of the wing segments, the Ultimaker Cura 4.10 software system (Ultimaker, Utrecht, The Netherlands) was used.

The manufacturing parameters for the carbon fiber filament (BASF Ultrafuse PAHT CF15) and glass fiber filament (Philament PLA Glass Reinforced) using the thermoplastic extrusion process for the UAV model components in Table 3 are presented.

The wing segments made from PAHT-CF15 filament were subjected to three-point bending tests and analyzed using finite element analysis (FEA). The maximum bending stress resulting from FEA was 66 MPa, and it was observed that both the tested structure and the finite element simulation exhibited the same mode of deformation, namely, local buckling of the skin in the region of applied load [74]. Additionally, within this study [74], when comparing experimental and simulated results of the wing segments and analyzing reaction forces, it can be stated that there was an error within a maximum range of 3%. The maximum three-point bending force for the wing segments, 3D-printed from PAHT-CF15 filament, reached a maximum value of 300 N with a maximum displacement of 19 mm.

**Table 3.** Manufacturing parameters for the UAV model components using the FFF process.

| FFF Parameter | Value | Value |
|---|---|---|
| Filament | BASF Ultrafuse PAHT CF15 | Philament PLA Glass Reinforced |
| Filament diameter [mm] | 2.85 | 2.85 |
| Layer height [mm] | 0.2 | 0.2 |
| Infill density [%] | 100 | 100 |
| Print speed [mm/s] | 45 | 50 |
| Travel speed [mm/s] | 100 | 80 |
| Printing temperature [°C] | 260 | 250 |
| Building plate temperature [°C] | 95 | 60 |
| Nozzle diameter [mm] | 0.6 | 0.6 |

*4.2. Additive Manufacturing of the UAV Fuselage*

For the additive manufacturing process of the UAV fuselage, it was divided into six sections during the design stage (Figure 9a) to accommodate the maximum 3D-printing volume of the Creat Bot DX-3D printer (Henan Creatbot Technology Limited, Zhengzhou City, China), which has dimensions of 300 mm × 250 mm × 520 mm. Before manufacturing the fuselage sections, specimens and components of the aircraft were tested and fabricated using different glass fiber filaments to evaluate their mechanical performance and determine the optimal manufacturing parameters for the FFF process [73]. To determine the mechanical performance (compression tests) of the fuselage sections, 3D-printed from PLA glass filament, two variants were tested: a fuselage section with longitudinal stringers and a fuselage section with X-shaped stringers oriented at ±45°. The 3D-printed fuselage sections with X-shaped stringers began to yield under a compressive force of 8.45 kN, while

those with straight stringers yielded at a load of 6.68 kN. It is evident that 3D printing the fuselage sections with stringers arranged at $\pm45°$ (with 100% infill density) results in structures with high rigidity compared to those with longitudinal stringers, which is why they are used in the fabrication of the UAV fuselage [73]. After the fuselage was segmented, each segment was exported from the SolidWorks software as an STL file to be loaded into the manufacturing preparation software, CreatBot 6.5.2 (Henan Creatbot Technology Limited, Zhengzhou City, Henan Province, China). The manufacturing procedure was the same for all fuselage segments: importing the STL model, generating the corresponding manufacturing program code, fabrication of the fuselage components (Figure 10a), removing the segments from the print bed, and removing the support material. During this stage, the most complex fuselage component, namely the central segment, was fabricated (Figure 10b), with a 3D-printing time of 74 h. The six fuselage segments manufactured using the FFF additive manufacturing process, with glass fiber filament, in Figure 10c are shown.

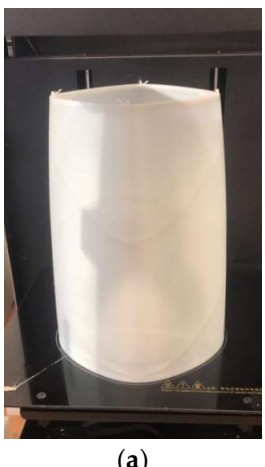
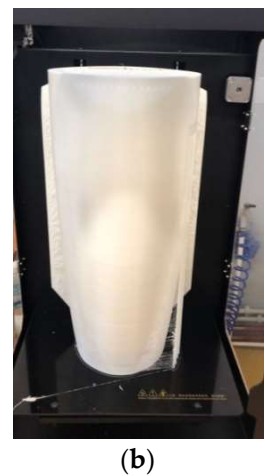
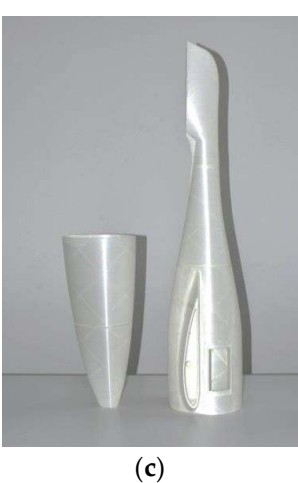

(**a**) (**b**) (**c**)

**Figure 10.** Additive manufacturing of the fuselage: (**a**) 3D printing of segment 3 of the fuselage; (**b**) 3D printing of segment 4 of the fuselage; (**c**) fuselage of the 3D-printed UAV model.

### 4.3. Additive Manufacturing of the UAV Empennage

Because of its dimensions, the stabilizer was divided in two components (Figure 11a) to fit within the maximum 3D-printing height of the printer. The UAV stabilizer was fabricated from carbon fiber filament (Figure 11a), while the elevator was made from glass fiber filament (Figure 11b). Significant difficulties in 3D printing were encountered during the printing process (frequent nozzle clogging, detachment of parts from the printing bed) for the composite filaments (glass fiber and carbon fiber), which are characteristic of the FFF process [91,92].

The additive manufacturing of the vertical empennage, consisting of the rudder and the vertical stabilizer, was carried out based on the 3D digital model, using glass fiber filament. The two main components of the vertical empennage (rudder and vertical stabilizer) were manufactured without being divided into sections, since they fit within the working volume of the Creat Bot DX-3D printer (Figure 11c). The two components (Figure 11d) were 3D-printed using support structures in the control lever area, where the control rods for the movable surface of the vertical empennage (rudder) will be attached.

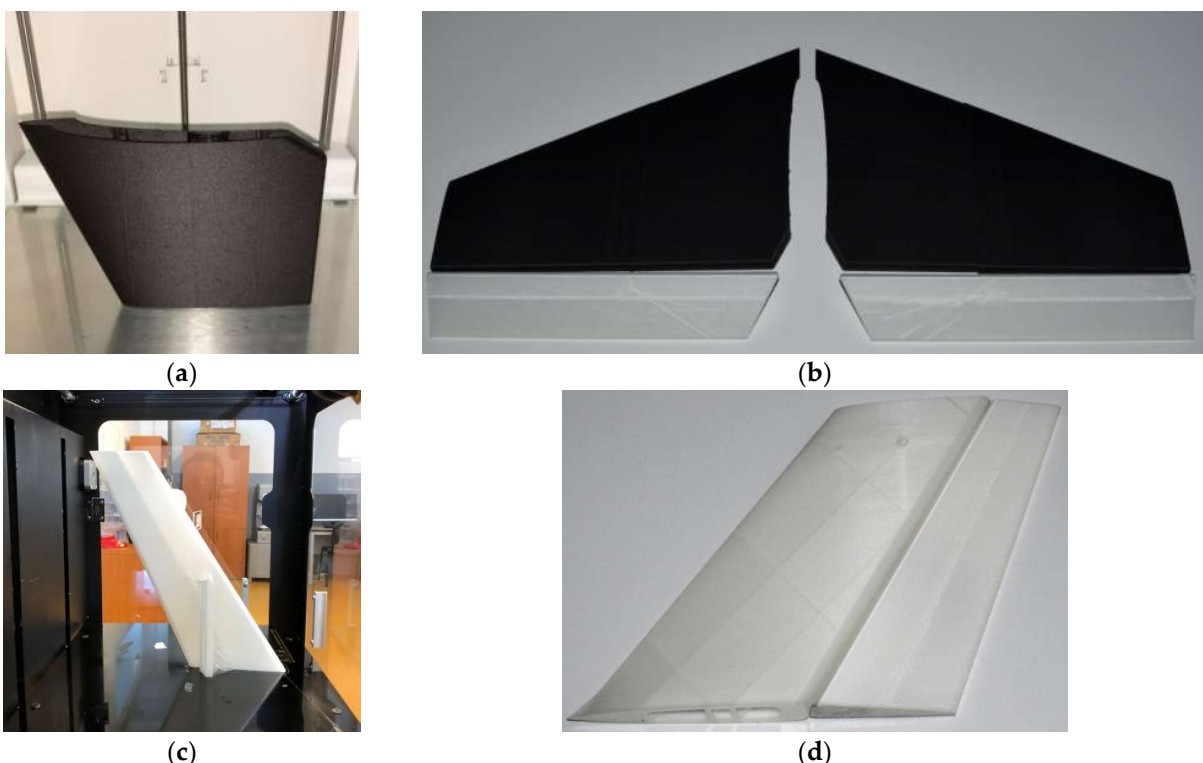

**Figure 11.** Additive manufacturing of the fuselage: (**a**) 3D printing of the stabilizer; (**b**) stabilizer—elevator assembly; (**c**) 3D printing of ruder; (**d**) vertical stabilizer—ruder assembly.

### 4.4. Additive Manufacturing of the UAV Landing Gear

Since the main landing gear is a vital element of the UAV model, its additive manufacturing and testing were carefully studied activities. Takeoff and landing are the most critical stages of flight, which requires a structural analysis of the landing gear. In this regard, the strength of the main landing gear was analyzed using four types of filaments (two with glass fiber and two with carbon fiber) as well as a landing gear manufactured using the vacuum-bagging process from high-quality carbon-fiber-reinforced polymer (CFRP), pressed into a polished mold. The four types of filaments used in the additive manufacturing of the landing gear were: polylactic acid matrix reinforced with short glass fibers (PLA GF), polypropylene matrix reinforced with short glass fibers (PP GF30), polyethylene terephthalate matrix reinforced with short carbon fibers (PET CF15), and polyamide matrix reinforced with short carbon fibers (PAHT CF15). For all four landing gear models, which were 3D-printed using the Ultimaker S5 3D printer, a 100% infill density was used [75]. Based on experimental tests (three-point bending) and finite element simulations, the following conclusion was drawn [75]: the use of landing gear manufactured by the FFF process is not suitable because this technology compromises the structural integrity of the entire UAV model during the most critical stages (takeoff and landing). Therefore, the main landing gear used for the UAV model is the one made of composite materials (CFRP), which exhibit significantly superior performance compared to models manufactured through additive technologies [75].

The finite element method can be employed from the early stages of UAV model development, as it provides rapid and cost-effective insights into the mechanical behavior of both the tested materials and the UAV model components [93,94]. Further studies on enhancing and optimizing the UAV model components and the materials used may be part of future research of the authors, potentially leading to the production of a second version of the UAV model.

## 5. Assembly of the UAV Model

### 5.1. Assembly of the UAV Model Components

The assembly of the UAV was performed on sub-assemblies: empennage, fuselage, and wing and landing gear. The main electronic components used in the assembly of the UAV model in Figure 12 are described.

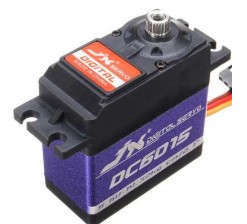 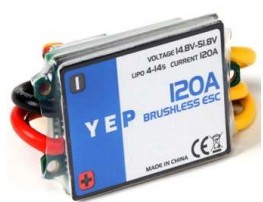 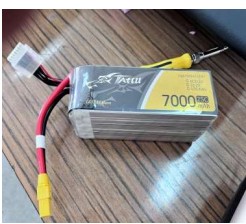

Weight: 62 g
Dimensions: 40.3 × 20.2 × 40.3 mm
Motor: Coreless
Gears: Metal
Voltage: 4.8–6.6 V
Stall Torque (6.6 V): 14.32 kg/cm
Speed (6.6 V): 0.1 s/60°

**(a)**

Weight: 20.3 g
Dimensions: 23 × 1 2 × 27.6 mm
Motor: Coreless
Gears: Metal
Voltage: 4.8–6 V
Stall Torque (6.6 V): 4.2 kg/cm
Speed (6.6 V): 0.08 s/60°

**(b)**

Weight: 180 g
Max Cont Current: 120 A
Voltage: 14.8 V–51.8 V
Input Voltage: 4–14 cells LiPo
Pulse width modulation: 8–16 KHz

**(c)**

Weight: 870 g
Capacity: 7000 mAh
Voltage: 22.2 V/6 Cell
Discharge Rate: 25 C
Size: 138 × 42 × 65 mm
Connector Type: XT-60

**(d)**

**Figure 12.** Electronic components used to assemble the UAV: (**a**) the servomechanism used to control the flaps and ailerons; (**b**) the servomechanism used to control the elevator and rudder; (**c**) brushless speed controller (ESC); (**d**) LiPo battery.

The assembly of the 3D-printed UAV components was carried out using medium density cyanoacrylate adhesive commonly used for bonding plastic/composite materials. The first stage involved the assembly of the empennages (vertical and horizontal). The assembly of the horizontal empennage began with positioning the carbon fiber rods in the central and leading-edge spars, followed by bonding the four segments of the stabilizer. For the elevator control, two servomechanisms were used (Figure 13a), placed between the two segments. Each servomechanism connection involved cable connections and checking the correct linkage of the elevator. A single servomechanism was used for the vertical empennage, positioned on the side surface of the rudder (Figure 13b), in the middle position. In this case, as well, the necessary cables were connected, and proper operation of the rudder was tested. The transmission of motion from the servomechanisms to the elevator and rudder was achieved through the coupling formed by forks connected using threaded rods.

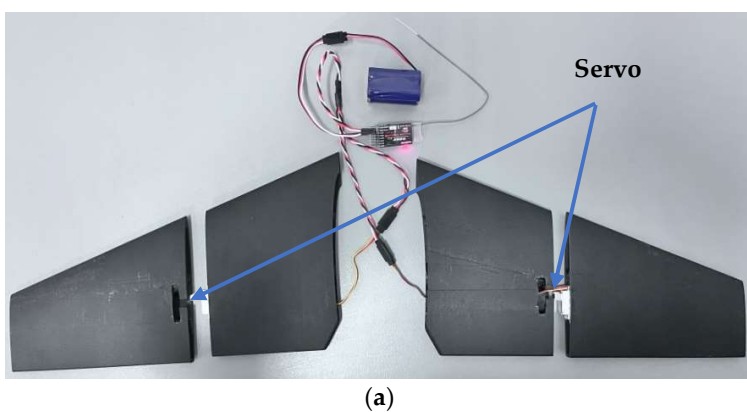 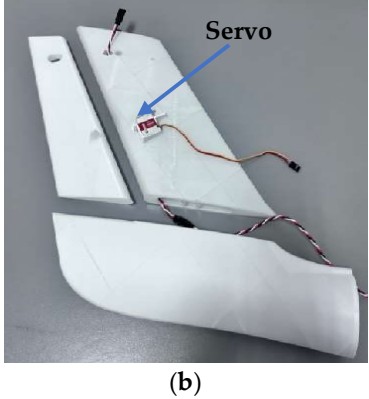

**(a)** **(b)**

**Figure 13.** Assembly of electronic components: (**a**) assembly and testing of the servomechanisms related to the horizontal empennage; (**b**) assembly and testing of servomechanisms related to vertical empennage.

The final step of empennage assembly was to assemble, through adhesive bonding, segment 6 of the fuselage (Figure 14a,b) to the fixed part of the vertical empennage (rudder). The assembly of the six sections of the fuselage started from the rear, by gluing the stabilizer to the rudder (Figure 14a).

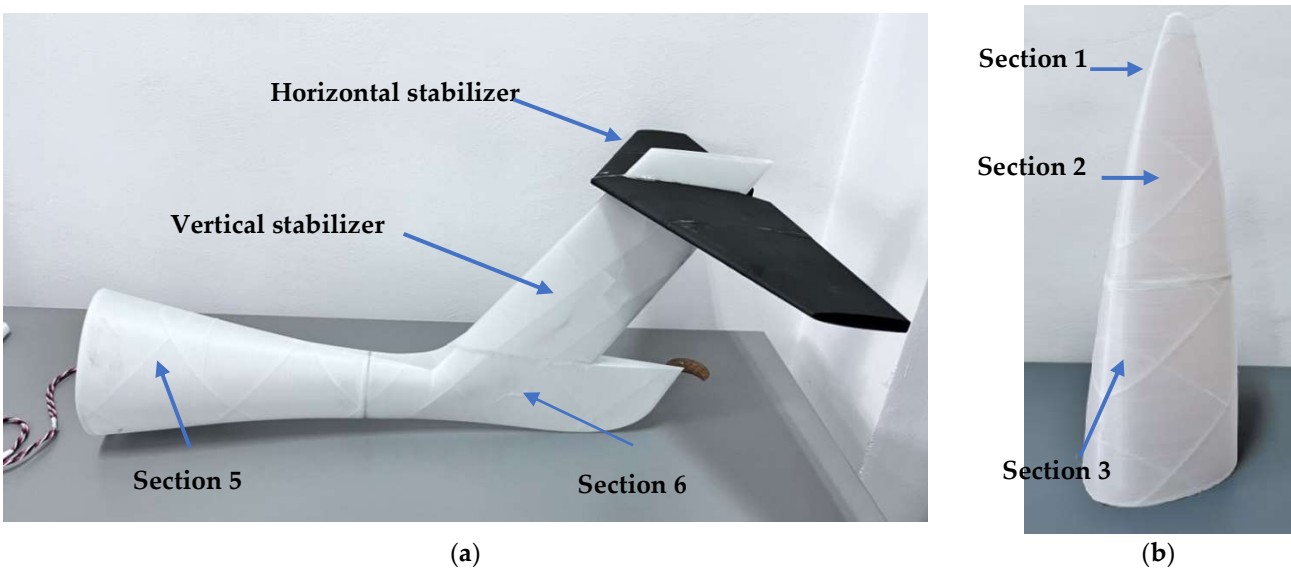

(a)

(b)

**Figure 14.** Assembling the fuselage sections: (**a**) initial assembly of the fuselage—rudder—stabilizer; (**b**) assembling the front sections.

For wing assembly, the gluing of the wing segments was carefully planned so that the last bonded segments (Figure 15a) would allow for the positioning of the control surfaces (flaps and ailerons) on the 3 mm diameter carbon rods. To achieve optimal alignment of the wing structure, the wing segment gluing was carried out with the carbon rod positioned in the cylindrical surface of the X-spar (Figure 15b). After the bonding stage of the main components (empennage, fuselage, wing), the next steps involved sanding the glued areas and painting the assembled components.

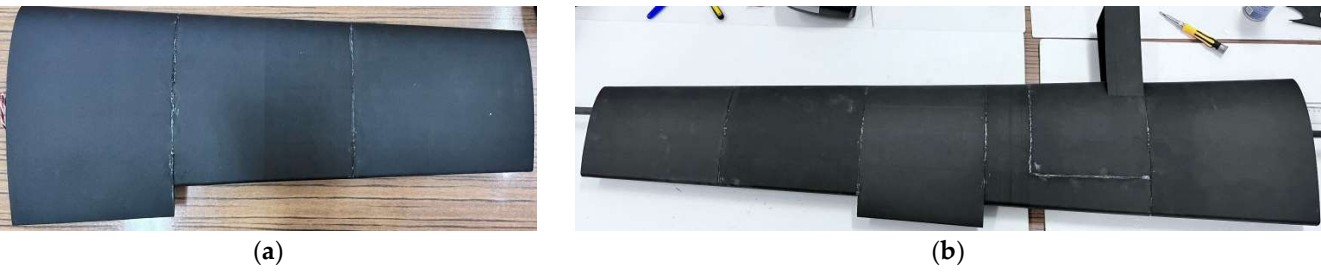

(a)

(b)

**Figure 15.** Assembling the UAV wing: (**a**) the first 3 sections; (**b**) fully assembled wing.

In the wing assembly, four servomechanisms were mounted inside the wing segments, with two servomechanisms operating the ailerons and two servomechanisms operating the flaps. After positioning the flaps and ailerons on the carbon rods, on which they rotate, the wing segments were glued together. The connection of the aileron and flap servomechanisms was carried out using control rods that link the servo arm and the component arm. After positioning the four control arms, the threaded rods were cut to ensure an appropriate turn, based on the specific functions they perform. The electric motors, manufactured through the SLS process [76], were assembled using four screws secured with nuts at the front side of the motor mount, with the help of two 2.5 mm carbon plates (Figure 16a).

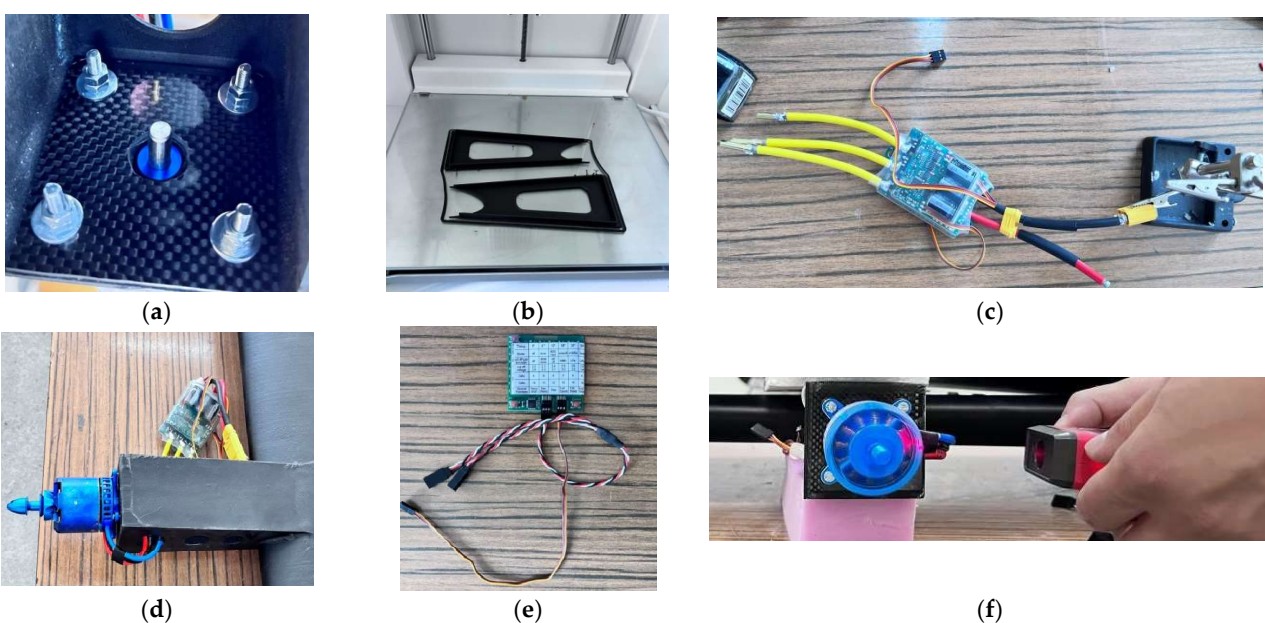

**Figure 16.** Assembly of electric motors: (**a**) assembly of the carbon plates for the motor support; (**b**) 3D printing of the motor support side covers; (**c**) bonding the speed controller connectors; (**d**) connecting the speed controller to the electric motor; (**e**) programming board for the speed controller; (**f**) verification of the rotation speed of the electric motors.

These carbon plates were designed to reinforce the front surface of the motor mount, which is one of the most stressed areas of the UAV. For the final assembly of the motor mount, the side covers with access cutouts for the motor-speed controller connection were 3D-printed using carbon fiber filament (Figure 16b). The next step involved soldering the speed controller connectors using solder (Figure 16c), which will, in turn, be connected to the electric motor connectors (Figure 16d). An important and complex stage was the configuration of the speed controller settings using the ESC programming card (Figure 16e). The main parameters set for the speed controller were the battery cell count and airplane mode, as well as motor-specific parameters and flight commands. After configuring the speed controller parameters, the last stage was to measure the rotational speed of the two electric motors, manufactured through the SLS process, using a tachometer (Figure 16f), resulting in equal values at various operating regimes.

*5.2. Testing the Electronic Systems of the UAV Model*

The GCSD4RSV2 ground control station (Digital Micro Devices, Valencia, Spain) serves as a central ground control system that allows human control over the unmanned aerial vehicle model. This professional portable ground control station is equipped with an intelligent antenna (with a control range of 150 km), radio control, telemetry (Mavlink and transparent data link), AES 128 encryption, a 10″ Full HD high-brightness IPS LED visible sunlight-readable display, 5.8 GHz video receiver, built-in PC with Windows 10, and a touchscreen. Regarding the transmission of telemetry data from the UAV model to the ground control station (GCS), it is achieved through a radio modem mounted on the UAV model. Using the Mavlink protocol, an open-source protocol, data are sent to the ground station through this modem. On the ground, the information is received by the PC through the USB or Bluetooth interface of the GCSD4RSV2 control station. This technology is compatible with most entry-level controllers such as the RXLRS system, Pixhawk flight controller, or APM. At the software level, everything is implemented on the Mission Planner software system, where takeoff and landing parameters, as well as flight routes, are set for the UAV model [95]. The UAV flight management system consists of three main subsystems: the ground control station, the communication link, and the

UAV model itself equipped with receivers and an autopilot [96]. The Pixhawk Orange cube allows the flight of various types of remote-controlled aircraft (helicopters, multirotor, airplanes), which turns it into a professional UAV platform.

The Pixhawk Orange cube features three accelerometers (one for each axis of the UAV aircraft), which require calibration. The combination of the transmitter assembly and antenna (SMBTS + BQ89) is called the intelligent antenna. Additionally, after verifying the operation of the ground control station, tests were conducted to assess the functionality of the FLIR Tau thermal module (FLIR Systems Inc., Boston, MA, USA) connected to the ground control station (Figure 17a,b).

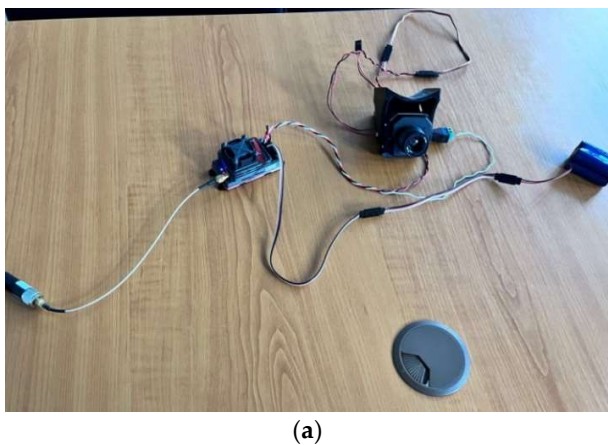
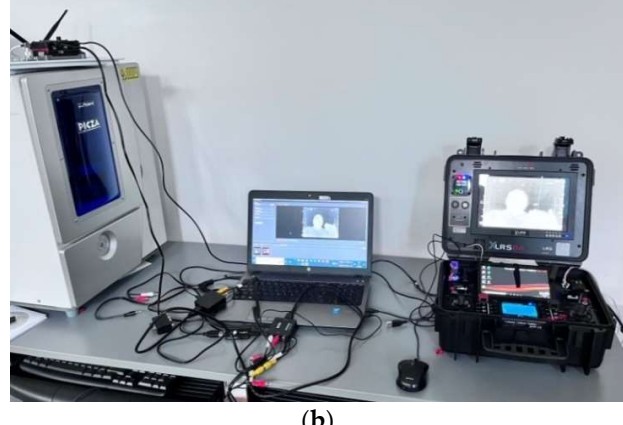

(**a**)     (**b**)

**Figure 17.** Testing the thermal module: (**a**) connecting the thermal module—smart antenna—battery—ground control station receiver; (**b**) image transmitted by the thermal module.

*5.3. Final Assembly of the UAV Model*

The final assembly of the UAV model began with the 3D printing, using fiberglass, of the two fairings that protect the motors and ESC system, while also ensuring efficient aerodynamic shapes. Prior to gluing the fairings, the propellers (16 × 8 inches) equipped with carbon fiber reinforcement were mounted. These propellers have a thin profile, low weight, and high efficiency. Since the initial setup of the electric motors and ESC programming, a contra-rotating mode has been utilized, which also applies to the propellers. During the assembly stage of the UAV model, the tail surfaces (horizontal and vertical empennages) were fabricated using thermoplastic extrusion. After assembling the tail surfaces' components (Figure 18a), the control surface positions (elevator and rudder) were determined and tested. For the assembly of the tail wheel landing gear (skid), a 3D-printed part was used to connect the UAV model's tail to the rear wheel assembly (Figure 18b). The rear wheel system consists of a metal rod, two springs, and an 8 cm diameter rubber wheel (Figure 18b). The two metal springs were installed to attenuate shocks during takeoff and landing phases of the UAV model. The main landing gear was attached to the fuselage of the UAV model using screw-nuts, with four screws in the central area (Figure 18c). To further dampen shocks, especially during takeoff and landing, the main landing gear was equipped with a hydraulic shock absorber and a 13 cm diameter rubber wheel. This hydraulic shock absorber was connected to the landing gear using a metal piece secured with four M4 screws (Figure 18d).

To secure the thermal module to the UAV model, a support was designed and manufactured using the thermoplastic extrusion process, which was then fastened to the fuselage using two screws (Figure 18e). After verifying the positioning of the servomotors for the control surfaces (ailerons and flaps), on the wing, the assembly of the left and right wing was checked. Their assembly was carried out through the upper part of the fuselage, using a mating system and a fixing screw (Figure 18f,g). Additionally, the carbon rods inside the wing were connected using aluminum alloy junctions with threaded surfaces (Figure 18g). Winglets were mounted on the carbon rod and then glued to the wingtip. An important

task was to establish, connect, and test all electronic connections (servomotors, electric motors, batteries, thermal module) to the receiver of the ground control station and the Pixhawk cube. Once all connections were verified, the fuselage was connected to the left wing and then to the right wing. Figure 18h illustrates the final structure of the UAV model equipped with the FLIR Tau thermal module.

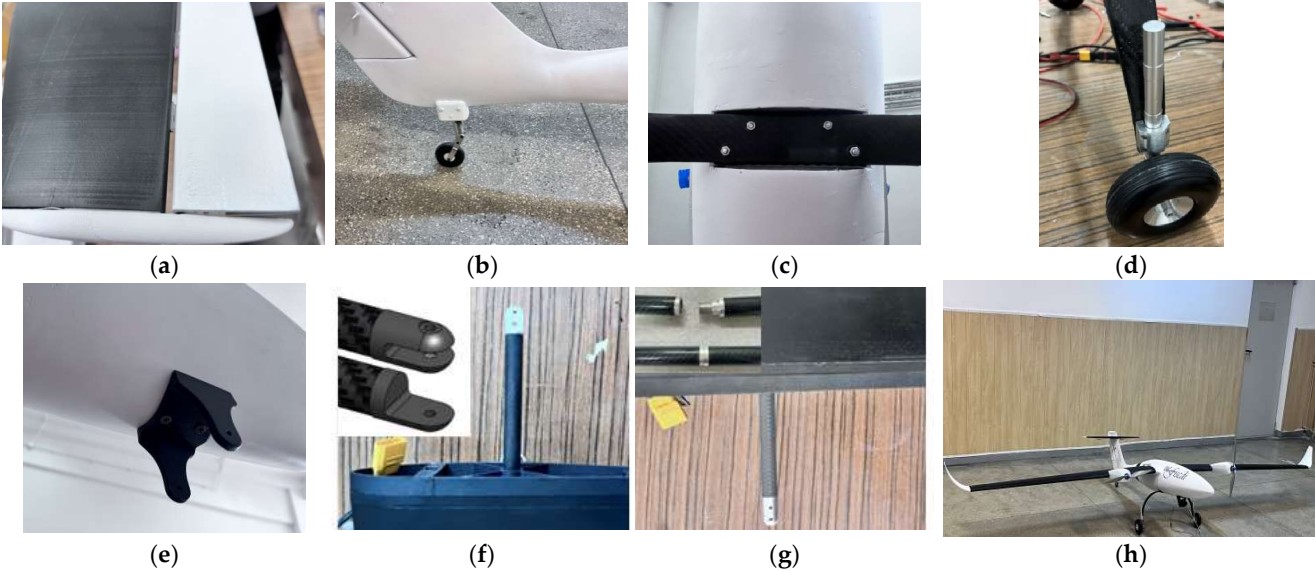

**Figure 18.** Assembly of the UAV model: (**a**) assembly of the stabilizer—elevator—carbon rod assembly and gluing of the horizontal empennage tips; (**b**) assembly of the rear landing gear; (**c**) attachment of the main landing gear; (**d**) assembly of the hydraulic shock absorber for the main landing gear and rubber wheel; (**e**) mounting of the support for the thermal module; (**f**) left wing half; (**g**) right wing half; (**h**) UAV model equipped with the thermal module.

Table 4 describes the weights and manufacturing time for the components of the UAV model made from composite materials.

**Table 4.** Weight of 3D-printed components of UAV composite model.

| Component | Weight [g] | Manufacturing Time [h] |
| --- | --- | --- |
| Fuselage | 1330 | 155 |
| Wings | 2680 | 310 |
| Ailerons | 170 | 28 |
| Flaps | 232 | 38 |
| Winglets | 240 | 36 |
| Horizontal tail | 248 | 52 |
| Vertical tail | 190 | 34 |
| Nacelle | 240 | 46 |
| Total | 5330 | 736 |

## 6. Testing and Verifying the Mission of the UAV Model

The testing of the model was carried out in two stages: the first stage involved ground testing of the UAV model in the workshop, and the second stage took place on a runway. The first stage began with a crucial activity, namely, the static balance of the model. As an experimental model, the balance was achieved by adding the two batteries (connected in parallel) in the nose area of the aircraft. This resulted in a balance point at 25% of the mean aerodynamic chord, ensuring a maneuverable aircraft with a quick response time.

After balancing the UAV aircraft, ground testing of the motors (Figure 19a) was conducted at maximum throttle. Within this stage, tests were also performed to ensure the connection between the UAV model, ground control station, Pixhawk Orange cube,

and the thermal module (Figure 19b). The runway testing of the UAV model consisted of the following (Figure 19c): verification of servo-mechanism operation, motor functionality, ground taxiing position, examination of the ground station connection with the aircraft, and verification of the thermal module's connection with the ground control station.

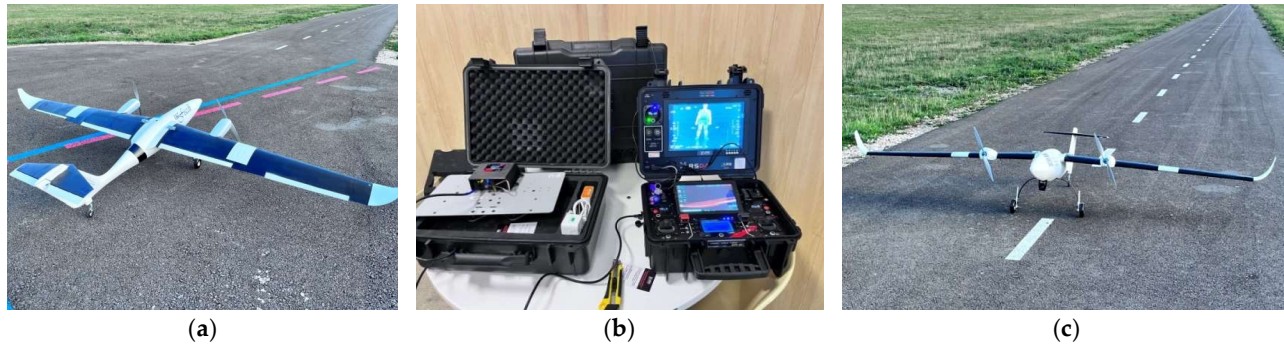

**Figure 19.** Ground testing of flight controls on the takeoff-landing runway: (**a**) motors testing; (**b**) thermal module testing; (**c**) preparation and verification of ground taxiing for the UAV model.

Following the flight testing, it was found that the UAV model, manufactured through the thermoplastic extrusion process, is stable and capable of achieving a wide range of speeds, with good aerodynamic characteristics and high maneuverability. For the flight tests, an asphalt location was chosen to allow for a rapid takeoff of the UAV model. The stages of the flight testing of the model were as follows: ground taxiing stage covering approximately 30–35 m, followed by the takeoff stage of the UAV model (Figure 20a), the third stage involved climbing to the flight ceiling (approximately 70 m), the horizontal flight stage (Figure 20b), mission execution (search and rescue), turning maneuvers, returning to the area of interest, and the final stage of landing the model on the ground.

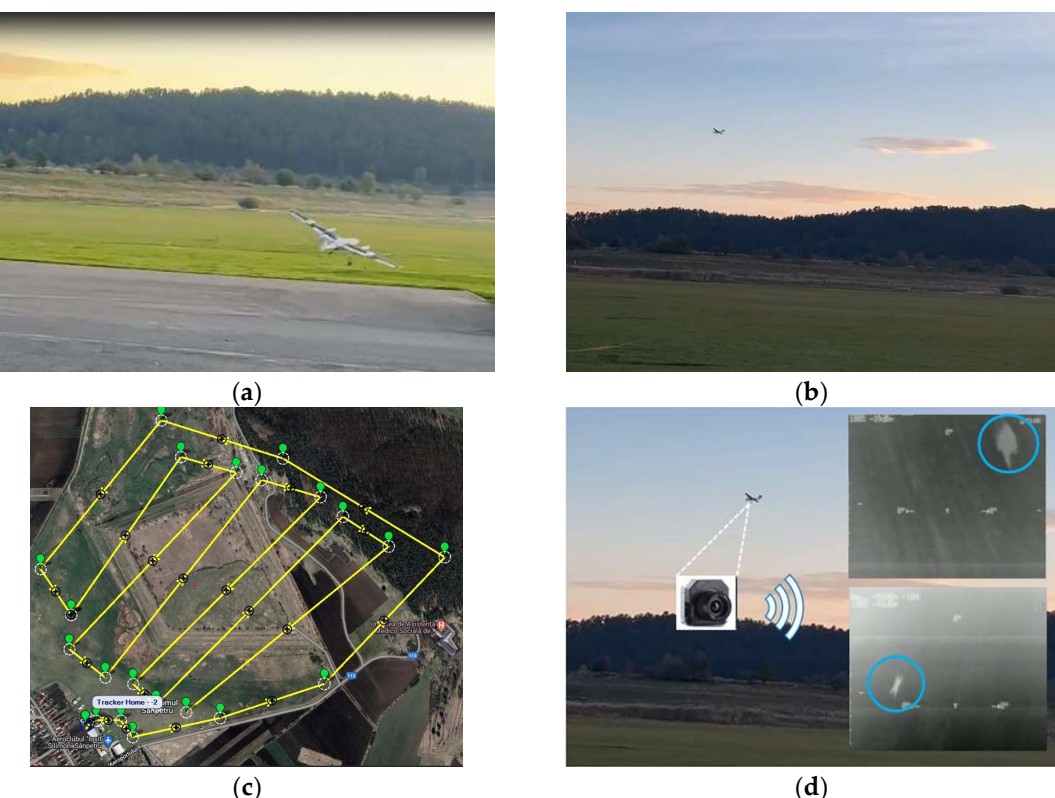

**Figure 20.** Flight of the UAV model: (**a**) takeoff; (**b**) cruising flight; (**c**) flight path establishment; (**d**) execution of the UAV model's flight mission using the thermal module.

Manual takeoff and landing were chosen for flight control, using the ground control station. Once the aircraft reached the predetermined flight ceiling, it was switched to semi-automatic mode by the control station, so that the aircraft can follow the predefined waypoints from the flight management program. The flight testing was successfully completed with safety measures in place and maximum maneuverability. The UAV model had a takeoff weight of 11 kg and autonomy of approximately 50 min. Using the map interface of the mission control software (Mission Planner, Q Ground Control), takeoff and landing profiles, as well as the flight path for the UAV model, were defined (Figure 20c) and then were loaded into the flash memory of the Pixhawk Cube Orange autopilot.

To fulfill the flight missions (search and rescue) of the UAV model using the thermal module and ground control station, images were captured during the horizontal flight phase. As a result, temperature variations were detected through the thermal module. In Figure 20d, a human (blue circle) being lying on the ground and another one moving were detected using body-temperature detection.

## 7. Conclusions

The paper presented a complete cycle of UAV model development, covering all necessary stages, such as: design, preliminary aerodynamic analysis, additive manufacturing, assembly, and flight testing. In the design phase, the dimensions of the UAV model were established, and then all structural components were described, starting from the wing to the landing gear. The final UAV model featured a configuration with the wing positioned on top of the fuselage, twin-engine, T-tail, and tricycle landing gear. The preliminary aerodynamic analysis highlighted that the unmanned aircraft exhibited a high maximum lift coefficient (approximately 1.2), classifying it as a powered glider UAV, with a large wing area and span. The FFF manufacturing of the UAV model's components was completed in approximately 736 h, using filament with short carbon fibers and short glass fibers. By utilizing the FFF process, complex components were manufactured at reduced costs and in a short timeframe for creating a UAV model. An advantage of FFF manufacturing for the aircraft components is the ability to intervene in the model's design at any time without adding additional costs, compared to traditional methods involving molds. The flight testing of UAV model was successfully completed with safety and maximum maneuverability. The UAV model had a takeoff mass of 11 kg, a wingspan of 3.4 m, an autonomy of approximately 50 min, and a control distance of about 100 km. By using the two brushless electric motors, manufactured through the SLS process, the UAV model had a wide range of speeds and was capable of performing search-and-rescue missions using the thermal module and ground control station. The missions that the UAV model can perform are to locate pilots and passengers after aviation accidents that may occur in hard-to-reach areas (such as mountains) and to estimate wild animal populations and detect illegal hunting. These missions can be successfully conducted using the real-time video transmission of the UAV model via the thermal module, which provides rescue teams with a clear and detailed view of the operational area, facilitating rapid and efficient responses.

In conclusion, this study has demonstrated the feasibility of creating the first unmanned aircraft, made from additive manufacturing composite materials, equipped with electric motors (manufactured through the SLS process), a traffic management system (ground control station and autopilot), and a thermal module for search-and-rescue missions. Therefore, the designed, analyzed, and manufactured UAV model serves as clear evidence that a nearly complete UAV model can be fabricated using the FFF process, capable of flying and operating with good aerodynamic characteristics and high maneuverability.

**Author Contributions:** Conceptualization, S.-M.Z., G.R.B., L.-A.C. and I.S.P.; methodology, S.-M.Z., M.A.P., L.-A.C. and V.M.S.; software, G.R.B. and C.L.; validation, S.-M.Z., G.R.B. and V.M.S.; investigation, S.-M.Z., M.A.P., G.R.B., V.M.S. and I.S.P.; writing—original draft preparation, S.-M.Z., G.R.B. and L.-A.C.; project administration, S.-M.Z. All authors have read and agreed to the published version of the manuscript.

**Funding:** This work was supported by a grant of the Ministry of Research, Innovation and Digitization, CNCS/CCCDI—UEFISCDI, project number PN-III-P2-2.1-PED-2019-0739, within PNCDI III.

**Data Availability Statement:** Not applicable.

**Acknowledgments:** We also acknowledge PRO-DD Structural Founds Project (POS-CCE, O.2.2.1., ID 123, SMIS 2637, ctr. No. 11/2009) for providing the infrastructure used in this work.

**Conflicts of Interest:** The authors declare no conflict of interest.

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
