# Peer review of "Material Extrusion Additive Manufacturing of the Composite UAV Used for Search-and-Rescue Missions"

_drones, doi:10.3390/drones7100602_

Round 1

Reviewer 1 Report

Comments and Suggestions for Authors

The paper describes the usage of additive manufacturing technology for the production of unmanned air vehicle. After a poor-quality introduction, there is a detailed design description which is very well written. Then, there is a part about the selection of all parts for the drone. The authors put some statements about the verification of produced parts but I cannot see any result of the research - it looks more like a report. There is a significant amount of issues that have to be significantly improved. I listed them below: 

1. The abstract part needs to be improved. Please read carefully the instructions for authors, it has to be max. 200 words long. Also in the abstract part there should be mentioned something about the novelty of your research. 

2. The whole introduction needs to be rewritten:
- you cannot use so many citations without and detailed literature review. Between lines 33-36 you put 19 citations without any discussion. 
- why did you compared AM materials to composites instead of conventionally made? 
- I cannot see any highlight of the novelty of your research. You need to describe the research area that you want to take care of. 

3. There is a lack of information about the reason for using that kind of UAV in your research. Please clarify it.

4. You wrote so many words about advanced materials for 3D printing and then you used PLA with glass fibers. In this regard, you need to rewrite the whole introduction to fit it into your research. Now it looks like two separate parts without any connection. If you claim that your choice of material was right you need to properly justify it. 

5. What is PAHT filament? You need to describe your materials better. 

6. Table 2 - provides a source of used parameters. You took it from a producer or investigated it? 

7. There is nothing about the anisotropy of mechanical properties of produced parts. Especially since you produced most of the parts along Z axis which affect the decrease of mechanical properties the most. 

8. I cannot see any data bout the mechanical properties of the produced parts. 

9. A wing was produced for 620 hours? It is almost one month. This value looks ridiculous. Why it took so much time? 

10. The conclusion part needs to be rewritten. Your conclusion must provide the most important outcomes of your work. Provide these outcomes point-by-point. Now it provides nothing. 

Reviewer 2 Report

Comments and Suggestions for Authors

The manuscript was good and have potential to publish after address major revision as follows:

1.      Line 32, Nothing truly unique in its current state. Because of the lack of a novel, the current submission looks to be a replication or modified work. The authors must describe their novel in detail. This work should be rejected owing to a major issue.

2.      Line 33-36, please develop this paragraph since too simple that explain several explample of UAVs application.

3.      Line 41-43, regarding composite materials, please giving additional updated relevant reference

4.      Line 83, the explanation was poor. If the authors heard about engineering design process? For giving the explanation of preliminary design please provide engineering design process. It would take 7 steps so it would consist of at least 7 solid paragraph explanation in every step.

5.      Line 99, what is the basis for this equation? Please giving more rationalisation.

6.      Figure 1-5 UAV model design not clearly seen, detail engineering drawing along with the section was needed.

7.      Line 209, it was good to refer NACA. But please provide the explanation of NACA best practice in the present study.

Comments on the Quality of English Language

-

Reviewer 3 Report

Comments and Suggestions for Authors

This paper is well written and well organized. It seems that the idea based on a Material Science conclusion project. The Priliminary design fro such UAV is not adequate for publication in MDPI as well as the Additive Manufacturing teqnique concerning composites is not clear. The methods for 3D printing UAVs is recently evolved but with very low fedility as it is very hard to predict the structure stiffness and strength uner loading conditions. Consequently, the paper didn't project the idea behind integrating 3D printing along with filaments of carbon fiber and fiber glass.

I Reject this paper for publication in its current format.

Comments on the Quality of English Language

 Minor editing of English language required

Reviewer 4 Report

Comments and Suggestions for Authors

The authors are considering new technology in the manufacture of drone components. They propose to use composite filaments. This study has been based on three-dimensional design, preliminary aerodynamic analysis and supported by the fabrication and assembly of thermoplastic extrusion composite components, flight testing and search-rescue performance of an unmanned aerial vehicle.

The content of the paper is interesting. The study subject is relevant. The presentation of the result is acceptable. The abstract reflects the paper context. Analysis of the problem state is acceptable, but should be include some recommendations about the areas of exploitation of such drones. For example, authors can provide this recommendation based on the reviews of drone applications:

- Li, Y., Liu, M., Jiang, D., Application of Unmanned Aerial Vehicles in Logistics: A Literature Review, Sustainability, 2022, 14(21), 14473

These recommendations can be introduced in the paper’s introduction or conclusion.

The paper is well illustrated and provides all necessary information about drone production in form of photos, pictures and tables.

Round 2

Reviewer 1 Report

Comments and Suggestions for Authors

The authors made significant improvements, but there are still some minor issues: 

Ad. 8. If you published the results already you can put it as self-citation - it is allowed. 

Ad.9. How about the fuselage? If two wings took 310h and 2.6 kgs how it is possible that twice lighter fuselage took 347h? 

After correction of the mentioned issues, the paper can be published. 

Reviewer 2 Report

Comments and Suggestions for Authors

Well effort to authors in the revision stage, but some comments still needed to addressed as follows:

1.      Line 36, even after revision, the novel of present study still not exist. I am recommending the authors to provide literature searching from thee main database, there are Scopus, web of science, and medline. With some keywords, provide the results in form of table and graphic than give the analysis from literature searching that the present study have unbreaking novel.

2.      Please explain potential further study performing computational simulation to improve materials performance. It brings several advantages such as lower cost and faster results compared to experimental test. Please provide the information along with relevant reference as follows: https://doi.org/10.3390/biomedicines11030951

3.      Some manua;l handling in engineering design from first concept, revised, into improvement until establish needs to shows since arguing how the design was planning not clearly seen.

Comments on the Quality of English Language

-

Reviewer 3 Report

Comments and Suggestions for Authors

Dear Authors.

Thanks for the didicated comments rebutal (Not from me), as the manuscript whas improved and got oriented to its objective. However, the paper doesn't reflect any scientefic achievement regarding the new amalgametion of  a 3D printing technique along with the comp osite carbon fiber structure.

In order to accept this paper for publication, some modification should be done. 

1- Design steps for the design consytaints of the UAV along with equations.

2- Design constaints and structural analysis and/or experimental results for the UAV printed structure.

It is not make scense that the whole manuscript has only one equation and two figures based on an undergraduate design software (XFLR5).

So my recommendations is to improve the highlighted points and a major revision is needed.

Round 3

Reviewer 3 Report

Comments and Suggestions for Authors

The manuscript was enhanced dramatically, thank tou for your effort, just few minor corrections needed

1- All citations should be revised as the numbers is not accurate as arranged

Ex. (Additive manufacturing processes by material extrusion have begun to be used for 99 the realization of UAV prototypes that are tested in flight [4237,4338]. Starting from 100 three-dimensional digital models to flying UAV prototypes, this process is completed in 101 a few days with minimal designer intervention.)

2- Figure 1 has a very bad resolution and also fig. 2 3 and .... please revise

3- Table 21 is not numbered as 21 it is Table 2, please revise. (21)

4-All CL and CD and all aerodynamic coefficients shouls be written as CL and CD

5- Figure 8, very bad resolution, Fig. 8b, pressure distribution should be shown with the legend of unit and to be clear for the values of pressure.

6- Fig. 12, it should be presented also tabulated for the specifications of the used servos and speed controllers and the motor.

7- Table 43. Weight of 3D printed components of UAV composite model should be Table 4, the manufacturing time for the fuelage and the total is wrong, please revise.

Citations from 20 to 23 is not appropriate please check the template.

Usefull papers to be cited for design process of UAVs

1- "Design and manufacture of umanned aerial vehicles (UAV) wing structure using composite materials" Planung und Bau einer Flügelstruktur für unbemannte Luftfahrzeuge (UAV) unter Verwendung von Kompositwerkstoffen DOI 10.1002/mawe.201500351

2- “Design and Production of small Tailless Unmanned Aerial Vehicle”, 15th International Conference on Applied Mechanics and Mechanical Engineering, MTC, May, 29 - 31, 2012.

3-Design and Manufacture of a Solar-Powered Unmanned Aerial Vehicle for Civilian Surveillance Missions  , https://doi.org/10.5028/jatm.v8i4.678
